# Data Distribution Valuation with Incentive Compatibility

## Abstract

Data valuation is a class of techniques for quantitatively assessing the value of data for applications like pricing in data marketplaces. Existing data valuation methods define a value for a dataset $D$. However, in many use cases, users are interested not only in the value of a dataset, but in the *distribution* from which the dataset was sampled. For example, consider a buyer trying to evaluate whether to purchase data from different vendors. The buyer may observe (and compare) only a small sample from each vendor prior to purchasing the data, to decide which vendor's data distribution is most useful to the buyer. The core question of this work is *how should we compare the values of data distributions from their samples?* Under a Huber model for statistical heterogeneity across vendors, we propose a maximum-mean discrepancy (MMD)-based valuation method which enables theoretically principled and actionable policies for comparing data distributions from samples. We show theoretically that our method achieves incentive-compatibility, thus incentivizing the data vendors to report their data truthfully. We demonstrate the efficacy of our proposed valuation method against several existing baselines, on multiple real-world datasets (e.g., network intrusion detection, credit card fraud detection) and downstream applications (classification, regression).

## 1 Introduction

*Data valuation* is a widely-studied practice of quantifying the value of data (Sim et al., 2022). Today, data valuation methods define a value for a dataset $D$, i.e., a fixed set of samples (Ghorbani & Zou, 2019; Jia et al., 2019). However, many emerging use cases require a data user to evaluate the quality not just a dataset, but the *distribution* from which the data was sampled. For example, data vendors (e.g., data markets like Datarade and Snowflake, financial data streams (Miller & Chin, 1996; Ntakaris et al., 2018), information security data ENISA (2010)) offer a preview in the form of a sample dataset to prospective buyers (Azcoitia & Laoutaris, 2022). Similarly, enterprises that sell access to generative models may offer a limited preview of its output distribution to prospective buyers (OpenAI, 2022). Buyers use these sample datasets to decide whether they wish to pay for a full dataset or data stream—i.e., access to the *data distribution*. More concretely, the buyers would compare between different data distributions (via their respective sample datasets) to determine and select the more valuable one.

In such applications, existing dataset valuation metrics are missing two components: They do not formalize the value of the underlying sampling distribution, nor do they provide a theoretically principled and actionable policy for comparing different sampling distributions based on the sample datasets. For instance, if a data buyer uses an existing data valuation metric $\nu$ that maps a dataset to a real-valued valuation to compare two datasets $A$ and $B$ drawn from distributions $P_A$ and $P_B$, respectively, the buyer may conclude that dataset $A$ is more valuable by observing that $\nu(A)$ is larger than $\nu(B)$, possibly by some margin. However, existing valuation metrics provide no guarantee that the underlying distribution $P_A$ is more valuable than $P_B$, making it difficult for the buyer to make the decision between $P_A$ or $P_B$. To draw a conclusion about the values of $P_A, P_B$ based on the values of $A, B$ requires both *a precise definition for the value of the sampling distribution (e.g., $P_A$), and its relationship to the value of the sample dataset (e.g., $A$).*

We consider a setting where a data buyer wishes to evaluate $n$ data vendors, each with their own dataset $D_i$ drawn i.i.d. from distribution $P_i$, where $i \in [n]$. The vendors' distributions are *heterogeneous*, i.e.,

the $P_i$'s can differ across vendors. Such distributional heterogeneity can arise from natural variations in data (Chen et al., 2016) or adversarial data corruption (Gu et al., 2019; Wang et al., 2023). Each vendor can choose to provide data $\tilde{D}_i$ (Chen et al., 2020), drawn i.i.d. from a different distribution $\tilde{P}_i$. We thus identify three technical and modeling challenges: *(i)* What is a suitable heterogeneity model that captures realistic data patterns? *(ii)* How should one define the value of a sampling distribution under a given heterogeneity model? *(iii)* How to account for the fact that a provided (sample) dataset $\tilde{D}_i$ may *not* be drawn i.i.d. from the vendor's actual data distribution $P_i$? In other words, can we find a valuation method that is *incentive compatible (IC)*, i.e., incentivizes vendors to truthfully report their data (Blum & Gölz, 2021; Chen et al., 2020)? The solutions to these challenges should lead to a theoretically principled and actionable policy for comparing (the values of) sampling distributions.

To address these three challenges, we make three key design choices.

*(a) Heterogeneity model.* We assume that each vendor's data distribution $P_i$ follows a Huber model (Huber, 1964), which is a mixture model of an unknown true distribution $P^*$ and an arbitrary outlier distribution $Q_i$. While the Huber model does not capture all kinds of statistical heterogeneity, mixture models are a reasonable model for many types of data heterogeneity observed in practice (Bonhomme & Manresa, 2015; Park et al., 2010). More importantly, the Huber model enables a direct analysis under our design choices *(b)* and *(c)* below. In contrast, prior efforts have *not* explicitly formalized heterogeneity and thus did not provide a precise analysis on how heterogeneity affects the value of data (Agussurja et al., 2022; Chen et al., 2020; Wei et al., 2021).

*(b) Value of a sampling distribution.* We use the negated *maximum mean discrepancy* (MMD) (Gretton et al., 2012) between a reference distribution $P^*$ and $P$ as the value of the sampling distribution $P$. Then, we leverage a (uniformly converging) MMD estimator (Gretton et al., 2012) to derive actionable policies for comparing sampling distributions with theoretical guarantees. In other words, a buyer can compare (the values of) sampling distributions $P_i, P_{i'}$ (from vendors $i, i'$) based on the respective samples $D_i, D_{i'}$ to determine which is more valuable, and by how much.

*(c) Valuation with incentive compatibility.* We approach this question by proving that our valuation method is *incentive compatible*, i.e., by ensuring that when a vendor mis-reports their data (e.g., by not sampling i.i.d. from $P_i$), it decreases $D_i$'s value. This is typically achieved by utilizing a *reference* dataset; in existing works, such a reference takes the form of a fixed validation dataset (Ghorbani & Zou, 2019; Just et al., 2023; Kwon & Zou, 2022; Schoch et al., 2022; Wu et al., 2022), a task distribution (Amiri et al., 2022), or an (implicit) access to the test data distribution $P^*$ (Agussurja et al., 2022; Chen et al., 2020). However, obtaining such reference datasets can be challenging because (1) different data vendors disagree on the choice of reference (Sim et al., 2022; Xu et al., 2021b), (2) such a reference may not be available *a priori* (Chen et al., 2020; Sim et al., 2020), or (3) dishonest vendors try to overfit to the reference (Amiri et al., 2022). We propose instead to use the aggregate distribution $P_N$, which is a weighted mixture of all data vendors' distributions (Sec. 3), as the reference in place of $P^*$. We provide an error guarantee based on design choices *(a)* and *(b)* to derive IC guarantees.

We highlight that these design choices are *not* isolated answers to each of the technical challenges separately, but rather collectively made to address these challenges (e.g., the analytic properties of both Huber and MMD are required in comparing sampling distributions, and also for IC). Our specific contributions are summarized as follows:

- We formalize the problem of data distribution valuation and connect it to common workflows in data marketplaces. We show why existing dataset valuation metrics do not solve this problem.
- We propose an MMD-based method for data distribution valuation. Under a Huber model of data heterogeneity, we derive actionable policies for comparing sampling distributions by comparing the observed sample datasets, with theoretical guarantees.
- We show how to achieve IC by using the aggregate distribution $P_N$ as a reference, thus relaxing the common assumption of requiring a known reference dataset or distribution.
- We demonstrate on real-world classification (e.g., network intrusion detection) and regression (e.g., income prediction) tasks that our method is effective in identifying the most valuable sampling distributions, against existing baselines. We also show the empirical implications of incentive compatibility.

## 2 RELATED WORK

Existing dataset valuation techniques fall roughly in 2 categories: those that assume a given reference dataset, and those that do not. We defer additional discussion to App. B due to space constraints.

**With a given reference.** Several existing methods require a *given reference* in the form of a validation set (e.g., (Ghorbani & Zou, 2019; Kwon & Zou, 2022)) or a baseline dataset (Amiri et al., 2022). Data Shapley (Ghorbani & Zou, 2019), Beta Shapley (Kwon & Zou, 2022) and Data Banzhaf (Wang & Jia, 2023) utilize the validation accuracy of a trained model as the value of the training data. Class-wise Shapley (Schoch et al., 2022) evaluates the effects of a dataset on the in-class and out-of-class validation accuracies. Both LAVA (Just et al., 2023) and DAVINZ (Wu et al., 2022) use a proxy for the validation performance of the training data as their value, instead of the actual validation performance, to be independent of the choice of downstream task ML model (Just et al., 2023) or to remove the need for model training (Wu et al., 2022). Differently, Amiri et al. (2022) assume that the buyer provides a baseline dataset as the reference to calculate a relevance score used to evaluate the vendor's data. Therefore, these methods *cannot* be applied without such a given reference, which can be difficult to obtain in practice (Chen et al., 2020; Sim et al., 2020). In contrast, our method can be applied without a given reference, by carefully constructing a reference (Sec. 5).

**Without a given reference.** To relax the assumption of a given reference, Chen et al. (2020); Tay et al. (2022); Wei et al. (2021) construct a reference from the data from all vendors. While the settings of (Chen et al., 2020; Tay et al., 2022; Wei et al., 2021) can include heterogeneity in the vendors' data, they do not explicitly formalize it and thus cannot precisely analyze its effects on data valuation. In contrast, our method, via the careful design choices of the Huber model (to formalize heterogeneity) and MMD (in the valuation function), provides a precise analysis on the effect of heterogeneity on the value of data (Eq. (2)). Furthermore, these methods did not provide theoretical guarantees on the error arising from using their constructed reference in place of the ground truth (i.e., $P^*$). In contrast, by exploiting Observation 1 in the setting of multiple vendors and the MMD-based valuation (Eq. (1)), we provide such theoretical guarantees (e.g., Proposition 2). In a different approach to relax the assumption of a given reference, Sim et al. (2020); Xu et al. (2021b) remove the dependence on a reference; as a result they can produce counterintuitive data values under heterogeneous data (experiments in Sec. 6). The closest related work to ours is (Tay et al., 2022), which adopts the $\text{MMD}^2$ as a valuation metric, primarily for computational reasons. However, this work does not consider data distribution valuation, nor does it describe how to compare (the values of) distributions, let alone with theoretical guarantees (including IC).[1] An additional comparative discussion with $\text{MMD}^2$ (including a summary of the empirical comparison) is in App. B.

## 3 MODEL, PRELIMINARIES AND PROBLEM STATEMENT

Each data vendor $i \in N := \{1, \ldots, n\}$ holds a dataset $D_i := \{z_{i,1}, \ldots, z_{i,m_i}\}$ of size $m_i$, with each data point $z_{i,j}$ i.i.d. sampled from the sampling distribution $P_i$ (Chen et al., 2022); abusing notation somewhat, we write $D_i \sim P_i$. We assume the existence of $P^*$, an unknown ground truth distribution from which we want to sample, known as test distribution (Ghorbani & Zou, 2019; Jia et al., 2019), true data distribution (Agussurja et al., 2022) or the task distribution (Amiri et al., 2022). We assume that each sampling distribution $P_i$ follows a Huber model (Huber, 1964), defined as follows: $P_i = (1 - \varepsilon_i)P^* + \varepsilon_i Q_i$ where $\varepsilon_i \in [0, 1)$ and $Q_i$ is a distribution that captures the heterogeneity of vendor $i$ (Gu et al., 2019). Define $D_N := \cup_{i \in N} D_i$ as the union of all the datasets and define $P_N$ as s.t. $D_N \sim P_N$ (derived in Observation 1). For notational simplicity, we omit the subscript $i$ and write $D, P$ instead of $D_i, P_i$, respectively, where it is clear.

**Why Huber.** We adopt the Huber model because (i) it is sufficiently general and can model various sources of heterogeneity (Chen et al., 2016; 2018); (ii) it satisfies a "convexity" property (i.e., a mixture of Huber models is also a Huber model), which proves useful with multiple data vendors:

**Observation 1.** Let $m_N := \sum_{i \in N} m_i$, $\omega_i := m_i/m_N$ as the relative size, and $P_N := \sum_{i \in N} \omega_i P_i$. Then, $P_N = (1 - \varepsilon_N)P^* + \varepsilon_N Q_N$ where $\varepsilon_N = \sum_{i \in N}(\omega_i \varepsilon_i)$ and $Q_N = \varepsilon_N^{-1} \sum_{i \in N}(\omega_i \varepsilon_i Q_i)$.

The aggregate distribution $P_N$ is a mixture of each $P_i$ weighted according to the relative size $\omega_i$, and importantly, $P_N$ is also a Huber model. This is crucial in the theoretical results in Sec. 5.

---

[1] Indeed, our proofs for IC do *not* apply to $\text{MMD}^2$, and it is unclear if $\text{MMD}^2$ can provide the same guarantees.

**Definition 1** ($\gamma$-Incentive Compatibility). Consider a vendor with actual sampling distribution $P$, who instead chooses to report data from $\tilde{P}$ where w.l.o.g., $d(P, P^*) < d(\tilde{P}, P^*)$. For $\gamma \geq 0$, a distribution valuation function $\Upsilon(P) \mapsto \mathbb{R}$ is $\gamma$-incentive compatible if $\Upsilon(P) \geq \Upsilon(\tilde{P}) - \gamma$.

A similar relaxation of exact IC is adopted by Balcan et al. (2019, Definition 2). Note that $\gamma = 0$ recovers exact IC (and $\gamma < 0$ directly implies exact IC but is overly strict).

**Problem statement.** Given two datasets $D \sim P$ and $D' \sim P'$, we seek a $\gamma$-incentive compatible (IC) distribution valuation function $\Upsilon(\cdot)$ and a dataset valuation function $\nu(\cdot)$ which enable a set of conditions under which to conclude that $\Upsilon(P) > \Upsilon(P')$, given only $\nu(D)$ and $\nu(D')$. In particular, the valuation function does *not* require access to the ground truth distribution $P^*$ as reference (otherwise it is immediate by Corollary 3 in App. A).

Existing methods cannot be easily applied to solve this problem. First, existing methods (e.g., (Ghorbani & Zou, 2019; Jia et al., 2019) do *not* provide a definition for $\Upsilon$, so $\Upsilon(P)$ is not well-defined; hence, they do not analyze the conditions under which $\Upsilon(P) > \Upsilon(P')$. Additionally, methods that explicitly require a given reference (e.g., (Kwon & Zou, 2022; Wu et al., 2022; Schoch et al., 2022)) *cannot* be applied here. For other methods, additional non-trivial assumptions (e.g., (Chen et al., 2020, Assumption 3.2), (Wei et al., 2021, Assumption 3.1)) are required, elaborated in App. B.

### 3.1 BACKGROUND ON MAXIMUM MEAN DISCREPANCY (MMD)

The MMD is an integral probability metric proposed as a test if two distributions are the same.

**Definition 2** (MMD, Gretton et al. (2012, Lemma 6)). For two distributions $P, P'$ and a fixed kernel $k$ (bounded by $K \in \mathbb{R}$), the *maximum mean discrepancy* (MMD) between $P, P'$ is

$$d(P, P') := [\mathbb{E}_{X, X' \sim P} k(X, X') - 2\mathbb{E}_{X \sim P, W \sim P'} k(X, W) + \mathbb{E}_{W, W' \sim P'} k(W, W')]^{1/2}.$$

In addition to satisfying the triangle inequality, which we use in later results, the MMD has a (biased) estimator when $D \sim P$ and $D' \sim P'$, and $|D| = m, |D'| = m'$ (Gretton et al., 2012, Eq. (6)):

$$\hat{d}(D, D') := \left[ \frac{1}{m^2} \sum_{x, x' \in D} k(x, x') - \frac{2}{mm'} \sum_{(x, w) \in D \times D'} k(x, w) + \frac{1}{m'^2} \sum_{w, w' \in D'} k(w, w') \right]^{1/2}.$$

## 4 MMD-BASED DATA DISTRIBUTION VALUATION

A valuation function should have the property that a dataset $D^* \sim P^*$ has (with high probability) the highest possible value, as $P^*$ is the true distribution. Under our Huber model, the actual sampling distribution $P$ of a data vendor may be a mixture containing an outlier distribution $Q$. Hence, $D \sim P$ is statistically different from $D^*$. The value of $P$ should therefore depend on its statistical similarity to $P^*$ (equiv. $D$ to $D^*$), suggesting a natural valuation method, specified as follows:

$$\Upsilon(P) := -d(P, P^*), \quad \nu(D) := -\hat{d}(D, D^*). \tag{1}$$

In other words, the value $\Upsilon(P)$ of a vendor's sampling distribution $P$ is defined as the negated MMD between $P$ and $P^*$, while the value $\nu(D)$ of its sample dataset $D$ is defined as the negated MMD estimate between $D$ and the reference dataset $D^*$.

The choice to use MMD in Eq. (1) has many benefits: (i) It is non-parametric and sample-based, making it appealing for data valuation. To elaborate, there is no need to explicitly learn the distributions $P, P'$ to obtain their distribution divergence (e.g., in Kullback-Leibler divergence-based valuations (Agussurja et al., 2022; Wei et al., 2021)), or resort to a surrogate in implementation (e.g., in Wassertein metric-based valuation (Just et al., 2023)). A more detailed comparison with other distances/divergences is in App. B. (ii) It can be applied to high-dimensional data by combining a kernel with a suitable feature transformation (Liu et al., 2020) or dimensionality reduction technique (e.g., principal component analysis) (Tay et al., 2022). (iii) It admits a direct and interpretable result under the Huber model on the quantitative effect of heterogeneity on the value of data (Eq. (2) below). Combining this with its known analytical properties, we derive theoretically principled and actionable policies (e.g., Proposition 1) for comparing data distributions.

**Effect of heterogeneity on data valuation.** Intuitively, the quality of a Huber distribution $P$ depends on both the size of the outlier component $\varepsilon$ and the statistical difference $d(P^*, Q)$ between $Q$ and $P^*$. A larger $\varepsilon$ and/or a larger $d(P^*, Q)$ decreases the value $\Upsilon(P)$. Our choice of valuation metric—specifically MMD—makes this intuition precise and interpretable. By Lemma 2, for $P = (1 - \varepsilon)P^* + \varepsilon Q$, we have that

$$\Upsilon(P) = -\varepsilon d(P^*, Q) \,. \tag{2}$$

Eq. (2) shows that for a fixed $\varepsilon$, $P$'s value decreases linearly w.r.t. $d(P^*, Q)$; similarly, for a fixed $d(P^*, Q)$, $P$'s value decreases linearly w.r.t. $\varepsilon$. Importantly, Eq. (2) enables subsequent results and a theoretically justified choice for the reference (e.g., Lemma 6 to derive Theorem 1 in Sec. 5).

**Data valuation with a ground truth reference.** With Eq. (1), we return to the problem statement described above: given two datasets $D \sim P$ and $D' \sim P'$, under what conditions can we conclude that $\Upsilon(P) > \Upsilon(P')$? We begin by assuming the access to a reference dataset $D^* \sim P^*$ from the true distribution, and relax this assumption in Sec. 5. Note that Corollary 3 (in App. A) states that having $D^* \sim P^*$ implies IC, so we focus on comparing $P, P'$ here.

**Proposition 1.** Given datasets $D \sim P$ and $D' \sim P'$, let $m := |D|$ and $m' := |D'|$. Let $D^* \sim P^*$ and $m^* := |D^*|$ be its size. For some bias requirement $\varepsilon_{\text{bias}} \geq 0$ and a required decision margin $\varepsilon_\Upsilon \geq 0$. If $\nu(D) > \nu(D') + \Delta_{\Upsilon,\nu}$ where the *criterion margin* $\Delta_{\Upsilon,\nu} := \varepsilon_\Upsilon + 2[\varepsilon_{\text{bias}} + \sqrt{K/m} + \sqrt{K/m'} + 2\sqrt{K/m^*}]$. Let $\delta := 2\exp(\frac{-\varepsilon_{\text{bias}}^2 \overline{m} m^*}{2K(\overline{m} + m^*)})$ where $\overline{m} = \max\{m, m'\}$. Then, $\Upsilon(P) > \Upsilon(P') + \varepsilon_\Upsilon$ with probability at least $(1 - 2\delta)$.

*(Proof in App. A)* Proposition 1 describes the criterion margin $\Delta_{\Upsilon,\nu}$ such that if $\nu(D) - \nu(D') > \Delta_{\Upsilon,\nu}$—i.e., the criterion is met—we can draw the conclusion that $\Upsilon(P) > \Upsilon(P') + \varepsilon_\Upsilon$ at a confidence level of $1 - 2\delta$. Hence, a smaller $\Delta_{\Upsilon,\nu}$ corresponds to an "easier" criterion to satisfy. The expression $\Delta_{\Upsilon,\nu} = \mathcal{O}(\varepsilon_\Upsilon + \varepsilon_{\text{bias}} + 1/\sqrt{\underline{m}})$ where $\underline{m} := \min\{m, m', m^*\}$ highlights three components that are in tension: a buyer-defined decision margin $\varepsilon_\Upsilon$, a bias requirement $\varepsilon_{\text{bias}}$ from the MMD estimator (Lemma 1), and the minimum size $\underline{m}$ of the vendors' sample datasets (assuming $m^* \geq \max\{m, m'\}$). If the buyer requires a higher decision margin $\varepsilon_\Upsilon$ (i.e., the buyer wants to determine if $P$ is more valuable than $P'$ by a larger margin), then it may be necessary to (i) set a lower bias requirement $\varepsilon_{\text{bias}}$ and/or (ii) request larger sample datasets from the vendors. In (i), suppose $\underline{m}$ remains unchanged, a lower $\varepsilon_{\text{bias}}$ reduces the confidence level $1 - 2\delta$ since $\delta$ increases as $\varepsilon_{\text{bias}}$ decreases. Hence, although the buyer concludes that $P$ is more valuable than $P'$ by a higher decision margin, the corresponding confidence level is lower. In (ii), suppose $\varepsilon_{\text{bias}}$ remains unchanged, a higher minimum sample size $\underline{m}$ increases the confidence level.[2] In other words, to satisfy the buyer's higher decision margin, the vendors need to provide larger sample datasets. This can also help the buyer increase their confidence level if the criterion is satisfied. Proposition 1 illustrates the interaction between a buyer and data vendors: The buyer's requirement is represented by the decision margin, and the vendors must provide sufficiently large sample datasets to satisfy this requirement.

## 5 A REFERENCE WITH BOUNDED ERROR FOR INCENTIVE COMPATIBILITY

Note that the previous section assumes access to reference distribution $P^*$. We now relax this assumption by replacing $P^*$ with the aggregate sampling distribution $P_N$ as the reference and derive an error guarantee to generalize Proposition 1 and the conditions for IC.

### 5.1 AGGREGATED SAMPLING DISTRIBUTION $P_N$ AS THE REFERENCE

Formally, using $P_N$ as the reference instead of $P^*$ gives the following valuation:

$$\hat{\Upsilon}(P) := -d(P, P_N) \,, \quad \hat{\nu}(D) := -\hat{d}(D, D_N) \text{ where } D_N \sim P_N \,. \tag{3}$$

Namely, $\hat{\Upsilon}$ is an approximation to $\Upsilon$ (equiv. $\hat{\nu}$ to $\nu$), with a bounded (approximation) error as follows,

**Proposition 2.** Recall $\varepsilon_N, Q_N$ from Observation 1. Then, $\forall P, |\Upsilon(P) - \hat{\Upsilon}(P)| \leq \varepsilon_N d(Q_N, P^*) \,.$

*(Proof in App. A)* Proposition 2 provides an error bound from using $P_N$ as the reference, which linearly depends on $\varepsilon_N$ and $Q_N$: A lower $\varepsilon_i$ (i.e., $P_i$ has a lower outlier probability) gives a lower

---

[2] In proof of Proposition 1, it is shown that the confidence level strictly increases when $m$ or $m'$ increases.

$\varepsilon_N$, and a lower $d(Q_i, P^*)$ (i.e., $P_i$'s outlier component is closer to $P^*$) leads to a lower $d(Q_N, P^*)$, resulting in a smaller error from using $P_N$ as the reference. We highlight that existing methods with a similar design to Eq. (1) (Chen et al., 2020; Tay et al., 2022; Wei et al., 2021) did not provide such an error guarantee (i.e., Proposition 2), which we use to generalize Proposition 1, next. Specifically, returning to our problem statement in Sec. 3 (solved earlier via Proposition 1 by assuming $P^*$), we relax the assumption of $P^*$ by adopting Eq. (3), instead of Eq. (1).

**Theorem 1.** Given datasets $D \sim P$ and $D' \sim P'$, let $m := |D|$ and $m' := |D'|$. Let $D_N$, $\hat{\nu}$ be from Eq. (3) and $m_N := |D_N|$. For some bias requirement $\varepsilon_{\text{bias}} \geq 0$ and a required decision margin $\varepsilon_\Upsilon \geq 0$, suppose $\hat{\nu}(D) > \hat{\nu}(D') + \Delta'_{\Upsilon,\nu}$ where the *criterion margin* $\Delta'_{\Upsilon,\nu} := \varepsilon_\Upsilon + 2\left[\varepsilon_{\text{bias}} + \sqrt{K/m} + \sqrt{K/m'} + 2\sqrt{K/m_N} + \varepsilon_N d(Q_N, P^*)\right]$. Let $\delta' := 2\exp(\frac{-\varepsilon_{\text{bias}}^2 \overline{m} m_N}{2K(\overline{m} + m_N)})$ where $\overline{m} = \max\{m, m'\}$. Then $\Upsilon(P) > \Upsilon(P') + \varepsilon_\Upsilon$ with probability at least $(1 - 2\delta')$.

*(Proof in App. A)* Compared with Proposition 1, the criterion margin $\Delta'_{\Upsilon,\nu}$ has an additional term of $2\varepsilon_N d(Q_N, P^*)$, which depends on both the size of the outlier component $\varepsilon_N$ and the statistical difference $d(Q_N, P^*)$ between $Q_N$ and $P^*$. This term explicitly accounts for the statistical difference $d(P_N, P^*)$ to generalize Proposition 1: $d(P_N, P^*) = 0$ recovers Proposition 1. Importantly, this result implies that using $P_N$ (to replace $P^*$) retains the previous analysis and interpretation: a buyer's requirement via the decision margin can be satisfied by the vendors providing (sufficiently) large sample datasets, which is empirically investigated in a comparison against existing valuation methods (App. C). We highlight that Theorem 1 exploits the convexity of Huber (via Observation 1), the triangle inequality of MMD (via Proposition 2) and the uniform convergence of the MMD estimator. Hence, the design choices of Huber and MMD are both necessary.

## 5.2 INCENTIVE COMPATIBILITY FOR TRUTHFUL REPORTING

Now we return to achieving IC without $P^*$ as the reference, by explicitly deriving the conditions for IC based on $P_N$ as the reference, namely for $\hat{\Upsilon}$ in Eq. (3).

**Corollary 1.** Let $P_i, \tilde{P}_i$ have relative size $\omega_i$. If $2d(P_{-i}, P^*) + d(P_i, P^*) - d(\tilde{P}_i, P^*) < 0$, then $\hat{\Upsilon}$ is exact IC; otherwise, $\hat{\Upsilon}$ is $\gamma_{P_N}$-IC with $\gamma_{P_N} = (1 - \omega_i)[2d(P_{-i}, P^*) + d(P_i, P^*) - d(\tilde{P}_i, P^*)]$.

*(Proof in App. A)* The assumption that $P_i, \tilde{P}_i$ have the same relative size $\omega_i$ is for the simplicity of the result, and is not necessary (see Proposition 3 in App. A). The if-condition describes how the two factors (i.e., $d(P_{-i}, P^*)$ vs. $d(P_i, P^*) - d(\tilde{P}_i, P^*)$) affect the IC of $\hat{\Upsilon}$. The term $d(P_{-i}, P^*) \geq 0$ quantifies the quality of the aggregate distribution of all vendors except $i$ as the reference (smaller is better). In general, $d(P_{-i}, P^*) > 0$ (otherwise $P^*$ is known and exact IC is implied by Corollary 3), so exact IC can be difficult to achieve. Informally, the degree of IC depends on the quality of the reference (i.e., $d(P_{-i}, P^*)$, and the severity of the mis-reporting (i.e., how negative $d(P_i, P^*) - d(\tilde{P}_i, P^*) < 0$ is) where a higher-quality reference and/or more severe mis-reporting implies a higher degree of IC (i.e., a lower $\gamma_{P_N}$ or even exact IC). Corollary 1 precisely characterizes the effect of these two factors, showing that exact IC is achieved for severe mis-reporting (i.e., if-condition is satisfied), and $\gamma_{P_N}$-IC is achieved otherwise.[3] By precisely characterizing the effect of mis-reporting on the value (via Eq. (2), we provide IC guarantees w.r.t. the actual value; in contrast, Chen et al. (2020, Definition 3.2) derive IC w.r.t. the expected value.

## 6 EMPIRICAL RESULTS

We empirically investigate the effectiveness of our method in ranking $n$ data distributions w.r.t. their performances, against existing baselines. Then, we verify IC under a specific mis-reporting implementation. Additional results under non-Huber setting, and for scalability are in App. C.

## 6.1 RANKING DATA DISTRIBUTIONS

Motivated by the use-case where a buyer wishes to identify the most valuable data vendor(s), we investigate how well our method (and existing baselines) correctly rank $n$ data vendors' distributions based on the values of the sample datasets.

---

[3]An additional discussion on the possible implications of the vendors' behaviors is in App. A.

**Setting.** For a valuation metric $\nu$ (e.g., Eq. (1)), denote the values of datasets from all vendors as $\boldsymbol{\nu} \coloneqq \{\nu(D_i)\}_{i \in N}$. To compare against different baselines (i.e., other definitions of $\nu$), we define the following *common* ground truth, the expected test performance $\zeta_i \coloneqq \mathbb{E}_{D_i \sim P_i}[\text{Perf}(\mathbf{M}(D_i); D_{\text{test}})]$ of an ML model $\mathbf{M}(D_i)$ trained on $D_i$, over a fixed test set $D_{\text{test}}$ where the expectation is over the randomness of $D_i$. Let $\boldsymbol{\zeta} \coloneqq \{\zeta_i\}_{i \in N}$. The ML model $\mathbf{M}$ and test set $D_{\text{test}}$ are specified below. Our implementation of MMD (including the radial basis function kernel) follows Li et al. (2017).

**Evaluation metric.** Here $\nu$ is effective if it identifies the most valuable sampling distribution, or more generally, if $\boldsymbol{\nu}$ preserves the ranking of $\boldsymbol{\zeta}$. In other words, the ranking of the data vendors' sampling distributions $\{P_i\}_{i \in N}$ is correctly identified by the values of their datasets $\{D_i\}_{i \in N}$, quantified via the Pearson correlation coefficient: $\text{pearson}(\boldsymbol{\nu}, \boldsymbol{\zeta})$ (higher is better). Note that we compare the rankings of different baselines instead of the actual data values which can be on different scales.

**Baselines.** To accommodate the existing methods which explicitly require a validation set (Sec. 2), we perform some experiments using a validation set $D_{\text{val}} \sim P^*$. Note that this assumption is difficult to guarantee in practice (Xu et al., 2021b), made only for empirical comparison, and subsequently relaxed. $D_{\text{val}}$ is different from $D_{\text{test}}$ (used to obtain $\zeta_i$ above). The baselines that explicitly require $D_{\text{val}}$ are class-wise Shapley (CS) (Schoch et al., 2022, Eq. (3)), LAVA (Just et al., 2023) and DAVINZ (Wu et al., 2022, Eq. (3)); the baselines that do not require $D_{\text{val}}$ are information-gain value (IG) (Sim et al., 2020, Eq. (1)), volume value (VV) (Xu et al., 2021b, Eq. (2)) and MMD$^2$ that leverages the square of MMD (Tay et al., 2022, Eq. (1)), which implements a biased estimator for MMD$^2$. We highlight that though theoretically MMD$^2$ is obtained by squaring MMD, the implemented estimator for MMD$^2$ is *not* obtained by squaring that for MMD. A more detailed discussion of Ours vs. MMD$^2$ is in App. B. For each baseline, we adopt their official implementation if available. We highlight that DAVINZ also includes the MMD as a specific implementation choice, linearly combined with a neural tangent kernel (NTK)-based score. However, their theoretical results are *not* specific to MMD while our application of MMD is theoretically justified (e.g., Theorem 1).

We also extend our method to explicitly consider the label information via the conditional distributions of labels given features (i.e., $P_{Y|X}$), denoted as Ours cond.[4] Specifically, for $D_i$ containing paired features and labels, we train a learner $\mathbf{M}(D_i)$ on $D_i$ and use its predictions on $D_{\text{val}}$ (thus not applicable without $D_{\text{val}}$) as an empirical representation of the $P_{Y|X}$ for $D_i$ and compute the MMD between the conditional distributions (more implementation details in App. C). It differs from Ours by exploiting the feature-label pairs in $D_i$. The assumption $D_{\text{val}} \sim P^*$ is relaxed by replacing $D_{\text{val}}$ with $D_N$, namely for the baselines that need an explicitly given reference, $D_N$ is the reference. The resulting data values are denoted as $\hat{\boldsymbol{\nu}}$ (the data values for when $D_{\text{val}}$ is available are denoted as $\boldsymbol{\nu}$).

**Datasets.** We consider both classification (Cla.) and regression (Reg.) datasets since some baselines (i.e., CS, LAVA) are specific to classification while some (i.e., IG, VV) are specific to regression. Note that our method is applicable to both. CaliH (resp. KingH) is a housing prices dataset in California (Kelley Pace & Barry, 1997) (resp. in Kings county Harlfoxem (2016)). Census15 (resp. Census17) is a personal income prediction dataset from the 2015 (resp. 2017) US census. (Muonneutrino, 2019). Credit7 (Narayanan, 2022) and Credit31 (Andrea Dal Pozzolo & Bontempi, 2015) are two credit card fraud detection datasets. TON (Moustafa, 2021) and UGR16 (Maciá-Fernández et al., 2018) are two network intrusion detection datasets. **Huber model.** A Huber model requires matched supports of $P^*$ and $Q$, such as the data/images of MNIST, EMNIST and FaMNIST in the same space $\mathbb{R}^{32 \times 32}$, similarly for CIFAR10 and CIFAR100, and Census15 and Census17. For other datasets, additional pre-processing is required: Both CaliH and KingH are standardized and pre-processed separately to be in $\mathbb{R}^{10}$. Additional pre-processing details are in App. C. Subsequently, each $P_i$ follows a Huber: $P_i = (1 - \varepsilon_i)P^* + Q$ (i.e., $\forall i, Q_i = Q$). The test set $D_{\text{test}}$ is from the respective $P^*$ and *not* seen by any data vendor. We also investigate some non-Huber settings where our method remains effective in App. C. **ML models.** For M, we adopt a 2-layer convolutional neural network (CNN) for MNIST, EMNIST, FaMNIST; ResNet-18 (He et al., 2016) for CIFAR10 and CIFAR100; logistic regression (LogReg) for Credit7 and Credit31, and TON and UGR16; linear regression (LR) for CaliH and KingH, and Census15 and Census17. Full details are in App. C.

**Results.** We report the average and standard error over 5 independent random trials, on CIFAR10/CIFAR100, TON/UGR16, CaliH/KingH and Census15/Census17 in Tables 1 to 4 respectively, and defer the others to App. C. Note that when $D_{\text{val}}$ is unavailable (i.e., right columns),

---

[4]Other baselines such as LAVA and CS already explicitly use the label information.

Ours cond. is not applicable because the label-feature pair information is not well-defined under the Huber model (e.g., for $P^* = $ CIFAR10 vs. $Q = $ CIFAR100, the label of a CIFAR100 image is not well-defined for a model trained for CIFAR10). The $\text{Perf}(\cdot)$ for $\zeta$ for classification (resp. regression) is accuracy (resp. coefficient of determination (COD)), so that a higher evaluation metric is better.

Table 1: **Cla.**: $P^* = $ CIFAR10, $Q = $ CIFAR100.

| Baselines | pearson$(\boldsymbol{\nu}, \boldsymbol{\zeta})$ | pearson$(\hat{\boldsymbol{\nu}}, \boldsymbol{\zeta})$ |
|---|---|---|
| LAVA | -0.907(0.01) | -0.924(0.01) |
| DAVINZ | -0.437(0.10) | -0.481(0.13) |
| CS | 0.889(0.03) | -0.874(0.02) |
| MMD$^2$ | 0.764(0.02) | 0.563(0.01) |
| Ours | 0.763(0.02) | **0.564(0.02)** |
| Ours cond. | **0.989(0.01)** | N.A. |

Table 2: **Cla.**: $P^* = $ TON, $Q = $ UGR16.

| Baselines | pearson$(\boldsymbol{\nu}, \boldsymbol{\zeta})$ | pearson$(\hat{\boldsymbol{\nu}}, \boldsymbol{\zeta})$ |
|---|---|---|
| LAVA | 0.254(0.26) | -0.159(0.38) |
| DAVINZ | -0.201(0.26) | -0.529(0.21) |
| CS | 0.451(0.19) | 0.256(0.28) |
| MMD$^2$ | 0.526 (0.11) | **0.480(0.15)** |
| Ours | **0.584(0.17)** | 0.461(0.14) |
| Ours cond. | 0.562(0.16) | N.A. |

For classification, Tables 1 and 2 show that our method performs well when $D_{\text{val}}$ is available (e.g., Ours cond. is the highest in the left column of Table 1) and also when $D_{\text{val}}$ is unavailable (e.g., Ours as highest in the right column of Table 1). MMD$^2$ performs comparably to Ours, which is expected since in theory their values differ only by a square and the evaluation mainly focuses on the rank, instead of the absolute values. We also note that CS, by exploiting the label information in classification, performs competitively with $D_{\text{val}}$, but performs sub-optimally without $D_{\text{val}}$. This is because the label information in $D_{\text{val}}$ is no longer available in $D_N$ (due to $D_N$ being Huber). LAVA and DAVINZ, both exploiting the gradients of the ML model, do not perform well.[5] The reason could be that under the Huber model, the gradients are not as informative about the values of the data. Intuitively, while the gradient of (the loss of) a data point on an ML model can be informative about the value of this data point, this reasoning is not applicable here, because the data point may not be from the same true distribution $P^*$: The value of a gradient obtained on a CIFAR100 image to an ML model intended for CIFAR10 may not be informative about the value of this CIFAR100 image. We highlight that neither baseline was originally proposed for such cases (i.e., the Huber model).

Table 3: **Reg.**: $P^* = $ CaliH, $Q = $ KingH.

| Baselines | pearson$(\boldsymbol{\nu}, \boldsymbol{\zeta})$ | pearson$(\hat{\boldsymbol{\nu}}, \boldsymbol{\zeta})$ |
|---|---|---|
| IG | -0.907(0.02) | |
| VV | -0.603(0.01) | |
| DAVINZ | 0.852(0.03) | 0.048(0.08) |
| MMD$^2$ | 0.872(0.03) | 0.726(0.09) |
| Ours | **0.896(0.02)** | **0.767(0.04)** |
| Ours cond. | 0.812(0.02) | N.A. |

Table 4: **Reg.**: $P^* = $ Census15, $Q = $ Census17.

| Baselines | pearson$(\boldsymbol{\nu}, \boldsymbol{\zeta})$ | pearson$(\hat{\boldsymbol{\nu}}, \boldsymbol{\zeta})$ |
|---|---|---|
| IG | -0.932(0.02) | |
| VV | -0.707(0.01) | |
| DAVINZ | 0.779(0.14) | 0.227(0.11) |
| MMD$^2$ | **0.889(0.05)** | **0.838(0.08)** |
| Ours | 0.843(0.03) | 0.769(0.08) |
| Ours cond. | 0.848(0.06) | N.A. |

For regression, Tables 3 and 4 show that Ours and MMD$^2$ continue to perform well while baselines (i.e., IG and VV) that completely remove the reference perform poorly, as they cannot account for the statistical heterogeneity without a reference. Notably, DAVINZ performs competitively for when $D_{\text{val}}$ is available, due to its implementation utilizing a linear combination of an NTK-based score (i.e., gradient information) and MMD (similar to Ours), via an auto-tuned weight between the two. We find that for classification, the NTK-based score is dominant while for regression (and available $D_{\text{val}}$) the MMD is dominant. This could be because the models are more complex for the classification tasks (e.g., ResNet-18) as compared to linear regression models for regression, so the obtained gradients are more significant (i.e., higher numerical NTK-based scores). Thus, for regression, DAVINZ produces values similar to Ours, hence the similar performance. We highlight that DAVINZ focuses on the use of NTK w.r.t. a given reference, while our method focuses on MMD *without* such a reference, as evidenced by Ours outperforming DAVINZ without $D_{\text{val}}$ (i.e., right columns of Tables 3 and 4).

---

[5]There is a caveat with DAVINZ, explained next with the results for regression.

## 6.2 INCENTIVE COMPATIBILITY

We empirically demonstrate that mis-reporting decreases a vendor's value, thus verifying IC.

**Settings.** Vendor $i' \in N$ is designated to mis-report: $\tilde{D}_{i'} \leftarrow D_{i'} + \mathcal{N}(0, \mathbb{I}\sigma^2)$ for $\sigma^2 = 0.2$, namely, vendor $i'$ adds zero-mean Gaussian noise to the features of the data in $D_{i'}$. This ensures $d(\tilde{P}_{i'}, P^*) > d(P_{i'}, P^*)$ as in Definition 1. For evaluation, we compute the data values (i) using Eq. (1) with a test set $D^* \sim P^*$ as the reference, denoted as the ground truth (GT); (ii) using Eq. (3) (i.e., Ours) and Tay et al. (2022, Eq. (1)) (i.e., MMD$^2$), respectively, with $D_N$ as the reference. We include MMD$^2$ to investigate how the square affects IC (since IC of Ours requires the triangle inequality of MMD that is *not* satisfied by MMD$^2$). By Corollary 3 in App. A, using $D^*$ achieves IC, namely GT will correctly reflect the values of the vendors (including the mis-reporting $i'$). Hence, for Ours and MMD$^2$, IC is satisfied if the data values are consistent with GT.

**Results.** Fig. 1 (resp. Fig. 2) plots the average and standard error over 5 independent trials of the *change* in data values of the $n = 5$ (resp. $n = 10$) vendors as we sweep the identity of the mis-reporting vendor $i' \in \{2, 3, 4\}$.[6] Specifically, the change in data value (i.e., $y$-axis) is defined as the difference between the value of $i$ when some $i'$ (possibly $i' = i$) is mis-reporting, and that of $i$ for when no vendor is mis-reporting. IC implies a negative change (i.e., decrease) in value for the mis-reporting $i'$, which is observed for GT, Ours, and MMD$^2$ in both Figs. 1 and 2. Note that the magnitude of the decrease for mis-reporting vendor $i'$ under Ours is more significant than that under MMD$^2$; hence it may be easier to identify the mis-reporting vendor from the truthful ones. This happens because the MMD is bounded by 1 (due to the choice of RBF kernel), and the square operation makes the value for MMD$^2$ strictly smaller in magnitude than that for MMD (i.e., Ours), as $\forall x \in (0, 1), x^2 < x$ . Therefore, MMD may be empirically more appealing than MMD$^2$, in addition to satisfying a theoretical IC guarantee (e.g., Corollary 1), which may not be satisfied for MMD$^2$.

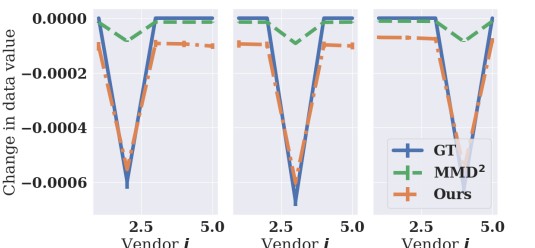 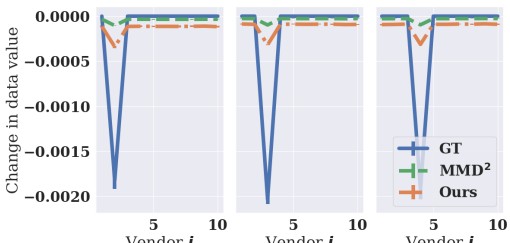

Figure 1: Change in data values for $P_{\text{MNIST}}$ and $Q_{\text{EMNIST}}$ with $n = 5$.

Figure 2: Change in data values for $P_{\text{MNIST}}$ and $Q_{\text{FaMNIST}}$ with $n = 10$.

## 7 DISCUSSION AND LIMITATIONS

Under a Huber model to formalize the heterogeneity of (the distributions of) data vendors, we propose a maximum mean discrepancy (MMD)-based data distribution valuation and derive theoretically principled and actionable policies for comparing two distributions from their respective samples. To address the practical constraint of not having a given reference, we propose to use the aggregate distribution as the reference and derive the corresponding error guarantee, which we exploit to derive incentive compatibility (IC). In particular, our IC guarantees elucidate the importance of the quality of each vendor's data, in incentivizing truthful reporting from others. Empirical results demonstrate that our method performs well in identifying the most valuable data vendor and achieving IC. One limitation is that exact IC is not always achieved, due to not knowing the ground truth distribution. Nevertheless, our method is empirically observed to satisfy IC. Another limitation is that the theoretical results are specific to the Huber model, but our method is observed to perform competitively under non-Huber settings. Hence, it is an interesting to extend to more general settings.

---

[6]The standard error bars are not visible because of the low variation across independent trials.

## REPRODUCIBILITY STATEMENT

We have included the necessary details to ensure the reproducibility of our theoretical and empirical results. Regarding theoretical results, the full set of assumptions, derivations and proofs for each theoretical result is clearly stated in either the main paper, or App. A. Regarding experiments: (i) the code to produce the experiments is included in a zip file as part of the supplementary material. It also contains the code and scripts to process the data used in the experiments. (ii) the processing steps and the licenses of the datasets used in the experiments, and the parameters (e.g., the choice of ML model used) that describe our experimental settings are clearly described in App. C. (iii) The information of the computational resources (i.e., hardware) used in our experiments and a set of scalability results for our method are included in App. C.

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

## A   PROOFS AND DERIVATIONS

### A.1   FOR SEC. 4

**Useful lemma.**

**Lemma 1** (Uniform Convergence of MMD Estimator (Gretton et al., 2012, Theorem 7)). Let $X \sim P, W \sim P'$ and the size of $X$ is $m$, the size of $W$ is $n$. Then the biased MMD estimator $\hat{d}$ satisfies the following approximation guarantee:

$$\Pr_{X,W} \left\{ |\hat{d}(X,W) - d(P,P')| > 2(\sqrt{K/m} + \sqrt{K/n}) + \varepsilon \right\} \leq 2\exp(\frac{-\varepsilon^2 mn}{2K(m+n)})$$

where $\Pr_{X,W}$ is over the randomness of the $m$-sample $X$ and $n$-sample $W$.

Note that $\varepsilon_{\text{bias}}$ in our results (e.g., Proposition 1) corresponds to $\varepsilon$ in Lemma 1.

**Proof of Proposition 1.**

*Proof of Proposition 1.* Apply Lemma 1 to $D \sim P$ and $D \sim P'$, respectively.

W.p. $\geq 1 - \delta_P$ where $\delta_P := 2\exp(\frac{-\varepsilon_{\text{bias}}^2 mm^*}{2K(m+m^*)})$,

$$d(P, P^*) \leq \hat{d}(D, D^*) + [2(\sqrt{\frac{K}{m}} + \sqrt{\frac{K}{m^*}}) + \varepsilon_{\text{bias}}]$$

$$-\Upsilon(P) \leq -\nu(D) + [2(\sqrt{\frac{K}{m}} + \sqrt{\frac{K}{m^*}}) + \varepsilon_{\text{bias}}]$$

$$\Upsilon(P) \geq \underbrace{\nu(D) - [2(\sqrt{\frac{K}{m}} + \sqrt{\frac{K}{m^*}}) + \varepsilon_{\text{bias}}]}_{A}$$

where the first inequality is from directly applying Lemma 1, the second inequality is from substituting the definitions in Eq. (1).

Symmetrically, w.p. $\geq 1 - \delta_{P'}$ where $\delta_{P'} := 2\exp(\frac{-\varepsilon_{\text{bias}}^2 m'm^*}{2K(m'+m^*)})$,

$$\Upsilon(P') \leq \underbrace{\nu(D') + [2(\sqrt{\frac{K}{m'}} + \sqrt{\frac{K}{m^*}}) + \varepsilon_{\text{bias}}]}_{B} .$$

Observe that if $A \geq B + \varepsilon_\Upsilon$, then apply the independence assumption (between $D \sim P$ and $D' \sim P'$), w.p. $\geq (1 - \delta_P)(1 - \delta_{P'})$, $\Upsilon(P) > \Upsilon(P') + \varepsilon_\Upsilon$ .

Re-arrange the terms in $A \geq B + \varepsilon_\Upsilon$ to derive $\zeta_\nu$,

$$\nu(D) - [2(\sqrt{\frac{K}{m}} + \sqrt{\frac{K}{m^*}}) + \varepsilon_{\text{bias}}] \geq \nu(D') + [2(\sqrt{\frac{K}{m'}} + \sqrt{\frac{K}{m^*}}) + \varepsilon_{\text{bias}}] + \varepsilon_\Upsilon$$

$$\nu(D) \geq \nu(D') + \underbrace{2[\varepsilon_{\text{bias}} + \sqrt{\frac{K}{m}} + \sqrt{\frac{K}{m'}} + 2\sqrt{\frac{K}{m^*}}]}_{\zeta_\nu} + \varepsilon_\Upsilon .$$

To arrive at the simpler but slightly looser result in the main paper. Note that

$$(1 - \delta_P)(1 - \delta_{P'}) \geq (1 - \delta)^2 \geq (1 - 2\delta) .$$

**Confidence level increases with $m, m'$.** Note that equivalently,

$$\delta_P = 2\exp(-\frac{\varepsilon_{\text{bias}}^2 m^*}{2K} + \frac{\varepsilon_{\text{bias}}^2 m^{*2}}{2K(m+m^*)}) ,$$

which is decreasing in $m$. Similarly for $\delta_{P'}$ w.r.t. $m'$. As a result, a higher $m$ implies a lower $\delta_P$ and thus a higher confidence level $(1 - \delta_P)(1 - \delta_{P'})$.

$\square$

### A.2 For Sec. 5

#### A.2.1 Useful lemmas - existing results

**Lemma 2** ((Chérief-Abdellatif & Alquier, 2022, In proof of Lemma 3.3)). For a Huber model $P := (1 - \varepsilon)P^* + \varepsilon Q$, the MMD $d(P, P^*) = \varepsilon d(P^*, Q)$.

#### A.2.2 Useful lemmas - our results

**Lemma 3.** Given a reference $P_{\text{ref}}$, denote $\varepsilon_{\text{ref}} := d(P_{\text{ref}}, P^*)$. Define $\hat{\Upsilon}(P) := -d(P, P_{\text{ref}})$ for a data distribution $P$. Then, $\hat{\Upsilon}(P) \in [\Upsilon(P) \pm \varepsilon_{\text{ref}}]$ .

*Proof of Lemma 3.* The proof follows directly by applying the triangle inequality of MMD $d(\cdot, \cdot)$ where the "triangle" is formed with $P^*$, $P$ and $P_{\text{ref}}$ and the definitions of $\Upsilon$ and $\hat{\Upsilon}$. □

In words, Lemma 3 provides the error range (via the upper and lower bounds) of $\hat{\Upsilon}(P)$ as an approximation to $\Upsilon(P)$ due to using $P_{\text{ref}}$ as the reference instead of $P^*$ (which is unknown in practice).

**A leave-one-out (LOO) perspective.** This LOO perspective refers to the paradigm in which data from all the vendors except $i$ are used to evaluate the data from $i$ (Chen et al., 2020). Specifically, the valuation of $P_i$ can be defined using $-d(P_i, P_{-i})$ (where $P_{-i} := m_{-i}^{-1} \sum_{i' \in N \setminus \{i\}} m_{i'} P_{i'}$ and $m_{-i} := \sum_{i' \in N \setminus \{i\}} m_{i'}$) instead of $-d(P_i, P_N)$. Due to the analytic properties of Huber and MMD, the LOO perspective can be equivalently derived via using the aggregate distribution $P_N$ as the reference (Lemma 4).

**Lemma 4.** For some $i \in N$, define $P_{-i} := m_{-i}^{-1} \sum_{i' \in N \setminus \{i\}} m_{i'} P_{i'}$ and $m_{-i} := \sum_{i' \in N \setminus \{i\}} m_{i'}$. Then,

$$\underbrace{d(P_i, P_N)}_{\text{ours}} = \frac{m_{-i}}{m_N} \underbrace{d(P_i, P_{-i})}_{\text{leave-one-out}} .$$

*Proof of Lemma 4.* Observe that

$$
\begin{aligned}
d(P_i, P_N) &\triangleq \|\mu_{P_i} - \mu_{P_N}\|_{\mathcal{H}_k} \\
&= \|\mu_{P_i} - (\frac{m_i}{m_N}\mu_{P_i} + \frac{1}{m_N} \sum_{j \in N \setminus \{i\}} m_{i'}\mu_{P_{i'}})\|_{\mathcal{H}_k} \\
&= \|(1 - \frac{m_i}{m_N})\mu_{P_i} - \frac{1}{m_N} \sum_{j \in N \setminus \{i\}} m_{i'}\mu_{P_{i'}}\|_{\mathcal{H}_k} \\
&= \|\frac{m_{-i}}{m_N}\mu_{P_i} - \frac{1}{m_N} \times m_{-i}\mu_{P_{-i}}\|_{\mathcal{H}_k} \\
&= \frac{m_{-i}}{m_N}\|\mu_{P_i} - \mu_{P_{-i}}\|_{\mathcal{H}_k} \\
&= \frac{m_{-i}}{m_N}d(P_i, P_{-i}) .
\end{aligned}
$$

Note that some steps require using the linearity of MMD $d(\cdot, \cdot)$ (Chérief-Abdellatif & Alquier, 2022, In proof of Lemma 3.3) as follows: suppose $P = (1 - \varepsilon)P^* + \varepsilon Q$, then,

$$
\begin{aligned}
d(P, P^*) &= \|(1 - \varepsilon)\mu_{P^*} + \varepsilon\mu_Q - \mu_{P^*}\|_{\mathcal{H}_k} \\
&= \|\varepsilon(\mu_Q - \mu_{P^*})\|_{\mathcal{H}_k} \\
&= \varepsilon\|\mu_Q - \mu_{P^*}\|_{\mathcal{H}_k} \\
&\triangleq \varepsilon d(Q, P^*) .
\end{aligned}
$$

The first and last lines are by definition of MMD, specifically its equivalent definition using the discrepancy w.r.t. the corresponding RKHS norm $\|\cdot\|_{\mathcal{H}_k}$ for the kernel $k$. The fourth line uses

the definition of $P_{-i} := \frac{1}{m_{-i}} \sum_{i' \in N \setminus \{i\}} m_{i'} P_{i'}$, which implies $m_{-i} P_{-i} = \sum_{i' \in N \setminus \{i\}} m_{i'} P_{i'}$. In particular, the equivalent mean-based representation for $P_{-i}$ in the RKHS (i.e., $\mu_{P_{-i}} := \sum_{i' \in N \setminus \{i\}} m_{i'} \mu_{P_{i'}}$) is used in the fourth line.

We highlight that the convexity of Huber is used in obtaining $P_N$ and $P_{-i}$, while the linearity of MMD is used in the derivations above. In other words, the above equivalence between $d(P_i, P_N)$ and $\frac{m_{-i}}{m_N} d(P_i, P_{-i})$ requires such analytic properties of Huber and MMD. $\square$

Importantly, this LOO perspective simplifies the analysis w.r.t. what a single vendor $i$ does. Plainly, due to this equivalence, we perform analysis using $d(P_i, P_{-i})$ instead of using $d(P_i, P_N)$. To elaborate, if vendor $i$ undertakes a unilateral change (e.g., changes $P_i$), then $d(P_i, P_{-i})$ is simpler to analyze since $P_{-i}$ is constant. On the other hand, $P_N$ changes if $P_i$ changes, so it is more difficult to analyse $d(P_i, P_N)$. Effectively, this LOO perspective "excludes" the effect of $P_i$ by using $P_{-i}$ instead of $P_N$ as the reference, thanks to the analytic properties of Huber and MMD.

By leveraging a leave-one-out perspective (Lemma 4), we can apply Lemma 3 to a specific reference $P_{\text{ref}} := P_N$ and obtain the upper and lower bounds of the value of a data vendor $i$:

**Corollary 2.** For the data distribution $P_i$ (of data vendor $i$), let $\hat{\Upsilon}(P_i) := -d(P_N, P_i)$.[7] Then,

$$\hat{\Upsilon}(P_i) \in \omega_{-i}(\Upsilon(P_i) \pm \xi_{-i})$$

where $\omega_{-i} := \frac{m_{-i}}{m_N}, m_{-i} := \sum_{i' \in N \setminus \{i\}} m_{i'}, \xi_{-i} := d(P_{-i}, P^*)$ and $P_{-i} := m_{-i}^{-1} \sum_{i' \in N \setminus \{i\}} m_{-i'} P_{-i'}$ is defined similarly to $P_N$ (in Observation 1) except w.r.t. $N \setminus \{i\}$ instead of $N$. In this way, $\omega_{-i}$ is the relative size of the sum of the samples from the vendors in the set $N \setminus \{i\}$.

*Proof.* Apply Lemma 4 to $\hat{\Upsilon}(P_i) := -d(P_i, P_N)$,

$$\hat{\Upsilon}(P_i) = -\omega_{-i} d(P_i, P_{-i}),$$

apply the triangle inequality of MMD to $d(P_i, P_{-i})$,

$$d(P_i, P^*) - d(P_{-i}, P^*) < d(P_i, P_{-i}) < d(P_i, P^*) + d(P_{-i}, P^*),$$

substitute the definition of $\xi_{-i} := d(P_{-i}, P^*)$,

$$d(P_i, P^*) - \xi_{-i} < d(P_i, P_{-i}) < d(P_i, P^*) + \xi_{-i},$$

multiply the entire (two inequalities) expression by $-\omega_{-i}$,

$$-\omega_{-i}(d(P_i, P^*) - \xi_{-i}) > -\omega_{-i} d(P_i, P_{-i}) > -\omega_{-i}(d(P_i, P^*) + \xi_{-i})$$
$$\omega_{-i}(-d(P_i, P^*) + \xi_{-i}) > -\omega_{-i} d(P_i, P_{-i}) > \omega_{-i}(-d(P_i, P^*) - \xi_{-i})$$
$$\omega_{-i}(\Upsilon(P_i) + \xi_{-i}) > -\omega_{-i} d(P_i, P_{-i}) > \omega_{-i}(\Upsilon(P_i) - \xi_{-i})$$
$$\underbrace{\omega_{-i}(\Upsilon(P_i) + \xi_{-i})}_{\hat{\bar{\Upsilon}}(P_i; \omega_i)} > \hat{\Upsilon}(P_i) > \underbrace{\omega_{-i}(\Upsilon(P_i) - \xi_{-i})}_{\hat{\underline{\Upsilon}}(P_i; \omega_i)}.$$

The first line directly multiplies the entire expression by $-\omega_{-i} < 0$; the third line applies the definition of $\Upsilon(P_i)$ and the last line applies the expression of $\hat{\Upsilon}$ above.

Note that, $\omega_{-i} = \frac{m_{-i}}{m_N}$ is used to denote the relative sizes for simplicity. The implication is that the actual sample sizes $\{m_i\}_{i \in N}$ are *not* necessary for the theoretical analysis, only the relative sizes $\{\omega_i\}_{i \in N}$ where each $\omega_i := \frac{m_i}{m_N}$. $\square$

### A.2.3 PROOFS AND DERIVATIONS OF MAIN RESULTS

**Proof of Observation 1.**

---

[7]Note that we overload the notation $\hat{\Upsilon}$ to specifically refer to using $P_N$ as the reference $P_{\text{ref}}$.

*Proof of Observation 1.* Note that $P_N$ is valid mixture model since $P_N$ is a convex combination of $P_i$'s. The expressions of $\varepsilon_N, Q_N$ are derived as follows,

$$P_N := \sum_i \frac{m_i}{m_N}[(1-\varepsilon_i)P^* + \varepsilon_i Q_i]$$

$$= P^* \sum_i \frac{m_i(1-\varepsilon_i)}{m_N} + (\sum_i \frac{m_i \varepsilon_i}{m_N} Q_i)$$

$$= P^*(1 - \underbrace{\sum_i \frac{m_i \varepsilon_i}{m_N}}_{\varepsilon_N}) + (\underbrace{\sum_i \frac{m_i \varepsilon_i}{m_N} Q_i}_{\varepsilon_N Q_N}) \ .$$

Then, the expression of $Q_N$ follows by applying the definition of relative size $\omega_i = \frac{m_i}{m_N}$. $\qquad\square$

**Proof of Theorem 1.**

*Proof of Theorem 1.* Apply Lemma 6 and Lemma 1 to $D \sim P$ and $D \sim P'$, respectively.

W.p. $\geq 1 - \delta'_P$ where $\delta'_P := 2\exp(\frac{-\varepsilon_{\text{bias}}^2 m m_N}{2K(m+m_N)})$,

$$\Upsilon(P) = -d(P, P^*) \geq -d(P, P_N) - d(P_N, P^*) \geq \underbrace{\hat{\nu}(D) - [2(\sqrt{\frac{K}{m}} + \sqrt{\frac{K}{m_N}}) + \varepsilon_{\text{bias}}] - \varepsilon_N d(Q_N, P^*)}_{A} \ ,$$

symmetrically, w.p. $\geq 1 - \delta'_{P'}$ where $\delta'_{P'} := 2\exp(\frac{-\varepsilon_{\text{bias}}^2 m' m_N}{2K(m'+m_N)})$,

$$\Upsilon(P') = -d(P', P^*) \leq -d(P', P_N) + d(P_N, P^*) \leq \underbrace{\hat{\nu}(D') + [2(\sqrt{\frac{K}{m'}} + \sqrt{\frac{K}{m_N}}) + \varepsilon_{\text{bias}}] + \varepsilon_N d(Q_N, P^*)}_{B} \ .$$

Observe that if $A \geq B + \varepsilon_\Upsilon$, and apply the independence assumption (between $D \sim P$ and $D' \sim P'$), then w.p. $\geq (1 - \delta'_P)(1 - \delta'_{P'})$, $\Upsilon(P) > \Upsilon(P') + \varepsilon_\Upsilon$ .

Re-arrange the terms in $A \geq B + \varepsilon_\Upsilon$ to derive $\zeta'_\nu$,

$$\hat{\nu}(D) - [2(\sqrt{\frac{K}{m}} + \sqrt{\frac{K}{m_N}}) + \varepsilon_{\text{bias}}] - \varepsilon_N d(Q_N, P^*) \geq \hat{\nu}(D') + [2(\sqrt{\frac{K}{m'}} + \sqrt{\frac{K}{m_N}}) + \varepsilon_{\text{bias}}] + \varepsilon_N d(Q_N, P^*) + \varepsilon_\Upsilon$$

$$\hat{\nu}(D) \geq \hat{\nu}(D') + 2\underbrace{\left[\varepsilon_{\text{bias}} + \sqrt{\frac{K}{m}} + \sqrt{\frac{K}{m'}} + 2\sqrt{\frac{K}{m_N}} + \varepsilon_N d(Q_N, P^*)\right]}_{\zeta'_\nu} + \varepsilon_\Upsilon \ .$$

To arrive at the simpler but slightly looser result in the main paper. Note that

$$(1 - \delta'_P)(1 - \delta'_{P'}) \geq (1 - \delta')^2 \geq (1 - 2\delta') \ .$$

$\qquad\square$

**Proof of Corollary 1.** Since Corollary 1 is a directly application of the more general case where $P_i, \tilde{P}_i$ having different relative sizes. We provide the result (i.e., Proposition 3) for this more general setting and then the proof of Corollary 1 directly follows.

**Proposition 3.** For fixed $P_i$ with relative size $\omega_i$, mis-reported $\tilde{P}_i$ with relative size $\omega'_i$, then $\hat{\Upsilon}$ from Eq. (3) satisfies the $\gamma'_{P_N}$-IC with $\gamma'_{P_N} = (\omega_{-i} + \omega'_{-i})d(P_{-i}, P^*) + \omega_{-i}d(P_i, P^*) - \omega'_{-i}d(\tilde{P}_i, P^*)$ .

*Proof of Proposition 3.* Apply Corollary 2 to $P_i, \omega_i$ to get its lower bound,

$$\hat{\Upsilon}(P_i) \geq \underline{\hat{\Upsilon}}(P_i, \omega_i) \ ,$$

which is equivalent to

$$\hat{\Upsilon}(P_i) - \underline{\hat{\Upsilon}}(P_i, \omega_i) \geq 0 .$$

Apply Corollary 2 to $\tilde{P}_i, \omega_i'$ to get its upper bound,

$$\hat{\bar{\Upsilon}}(\tilde{P}_i, \omega_i') \geq \hat{\Upsilon}(\tilde{P}_i) ,$$

which is equivalent to

$$0 \geq \hat{\Upsilon}(\tilde{P}_i) - \hat{\bar{\Upsilon}}(\tilde{P}_i, \omega_i') .$$

Then,

$$\hat{\Upsilon}(P_i) - \underline{\hat{\Upsilon}}(P_i, \omega_i) \geq \hat{\Upsilon}(\tilde{P}_i) - \hat{\bar{\Upsilon}}(\tilde{P}_i, \omega_i')$$
$$\hat{\Upsilon}(P_i) \geq \hat{\Upsilon}(\tilde{P}_i) - (\hat{\bar{\Upsilon}}(\tilde{P}_i, \omega_i') - \underline{\hat{\Upsilon}}(P_i, \omega_i))$$
$$\hat{\Upsilon}(P_i) \geq \hat{\Upsilon}(\tilde{P}_i) - (\omega_{-i}'(\Upsilon(\tilde{P}_i) + \xi_{-i}) - \omega_{-i}(\Upsilon(P_i) + \xi_{-i}))$$
$$\hat{\Upsilon}(P_i) \geq \hat{\Upsilon}(\tilde{P}_i) - \underbrace{(\omega_{-i}'(-d(\tilde{P}_i, P^*) + d(P_{-i}, P^*)) - \omega_{-i}(-d(P_i, P^*) + d(P_{-i}, P^*)))}_{\gamma_{P_N}'} .$$

The third line recalls the definitions of $\hat{\bar{\Upsilon}}(\tilde{P}_i, \omega_i')$ and $\underline{\hat{\Upsilon}}(P_i, \omega_i)$ from Corollary 2; the fourth line recalls the definitions of $\Upsilon(\cdot)$ from Eq. (1) and $\xi_{-i} = d(P_{-i}, P^*)$ (from Corollary 2). The proof is complete by rearranging the terms and noting the definition of $\gamma_{P_N}'$. □

*Proof of Corollary 1.* Follow from Proposition 3 by setting $\omega_i = \omega_i'$ and note that $\omega_{-i} = 1 - \omega_i$, $\hat{\Upsilon}$ satisfies $\gamma_{P_N}$-IC with $\gamma_{P_N} = (1 - \omega_i)[2d(P_{-i}, P^*) + d(P_i, P^*) - d(\tilde{P}_i, P^*)]$ .

In particular, recall that $\gamma_{P_N} < 0$ implies exact IC. Since $(1 - \omega_i) > 0$, hence, $\gamma_{P_N} < 0 \equiv 2d(P_{-i}, P^*) + d(P_i, P^*) - d(\tilde{P}_i, P^*) < 0$ implies exact IC. □

**Additional discussion on the implications of Corollary 1.**

We provide here a more detailed discussion and interpretation on the result (i.e., Corollary 1) itself, and also a discussion on the possible implications of the behaviors of the vendors, in light of this result.

As mentioned in the main text, this result characterizes how the interaction between two factors —the quality of the aggregate distribution of all vendors except $i$ (i.e., $d(P_{-i}, P^*)$) and the severity of $i$' mis-reporting (i.e., $d(P_i, P^*) - d(\tilde{P}_i, P^*) < 0$)— affects the degree of IC of $\hat{\Upsilon}$. In particular, when the if-condition is satisfied, $\hat{\Upsilon}$ achieves exact IC. This if-condition is satisfied when (i) the distributions of the vendors except $i$ are "good" (i.e., close to $P^*$ in MMD), so that their aggregate distribution $P_{-i}$ is also close to $P^*$, and/or (ii) vendor $i$ is severely mis-reporting (i.e., $d(P_i, P^*) - d(\tilde{P}_i, P^*) \ll 0$). In other words, $\hat{\Upsilon}$ directly disincentivizes a vendor $i$ from mis-reporting severely, under the condition that the distributions of the vendors except $i$ are close to $P^*$. However, when the if-condition is not satisfied and $\hat{\Upsilon}$ is only shown to achieve $\gamma_{P_N}$. The interpretation is that, $\hat{\Upsilon}$ is not shown to directly disincentivize vendor $i$ from mis-reporting "mildly" (i.e., $d(P_i, P^*) - d(\tilde{P}_i, P^*) < 0$ but its magnitude $|d(P_i, P^*) - d(\tilde{P}_i, P^*)|$ is small relative to $d(P_{-i}, P^*)$). The combined interpretation is that, $\hat{\Upsilon}$ directly incentivizes severe mis-reporting, but is not shown to directly disincentivize mild mis-reporting. The degree to which $\hat{\Upsilon}$ can disincentivize $i$ from mis-reporting depends on the quality of the aggregate distribution $P_{-i}$ of every vendor except $i$. We believe this result provides a useful way to relax the exact IC, by relaxing the stringent assumption of knowing the ground truth distribution $P^*$ (which implies exact IC by Corollary 3). In other words, exact IC is achieved by knowing $P^*$, but since $P^*$ is generally unknown in practice, we resort to a tractable alternative $P_N$, and obtain a corresponding result to guarantee a relaxed version of IC.

In light of this, one may ask: "Since there is a case that $\hat{\Upsilon}$ does not directly disincentivize a vendor $i$ from mis-reporting, what would the vendors do (e.g., mis-report or not, and how severely)? For instance, would a vendor mis-report, but only mildly?" We hypothesize two possible implications

(not necessarily related). First, if the vendor $i$ is the minority of the vendors who have a high-quality distribution (e.g., $i$ alone has access to $P^*$ while the distribution of every other vendor is different from $P^*$). Then, $i$ may not be disincentivized from mis-reporting, because if $i$ chooses to mis-report, it may not lead to a lower value (since Corollary 1 guarantees only $\gamma_{P_N}$-IC in this scenario). This is indeed observed in our experimental results (e.g., Table 12) when the "best" vendor chooses to mis-report and IC is empirically observed to be weaker. Hence, this vendor $i$ might choose to mis-report, which means not sharing the high-quality data with others. In a sense. $i$'s monopoly on good data affords $i$ an ability to mis-report without its value being decreased. This situation is thus undesirable to other vendors $i' \neq i$. To avoid such situations of a minority or even a monopoly on the good data by certain vendors, the (other) vendors may feel prompted to invest additional resources to improve their data collection processes (so as to improve their respective $P_i$'s), ideally encouraging a cycle of positive competition among the vendors. Second, note that the if-condition in Corollary 1 points out the importance of $P_{-i}$ in ensuring IC w.r.t. $i$. Then, in the eyes of $i$, choosing to report truthfully (instead of mis-reporting a worse $\tilde{P}_i$), can help improve the quality of $P_{-i'}$ (for some $i' \neq i$) and thus help to ensure IC w.r.t. $i'$. Intuitively, every vendor is part of a reference that could (help to) disincentivize every vendor from mis-reporting. Then, all the vendors reporting truthfully may be a weak equilibrium (though not theoretically shown yet and thus an interesting future direction), such as in (Chen et al., 2020).

**Corollary 3** (Knowing $P^*$ implies exact IC)**.** Suppose that $P^*$ is known and used in $\Upsilon$ for valuing the data vendors as in Eq. (1). Recall $\tilde{P}_i$ from Definition 1 as the mis-reported $P_i$. Then, $\Upsilon(P_i) > \Upsilon(\tilde{P}_i)$.

*Proof.* The proof follows directly by plugging the definition of $\Upsilon$ in Eq. (1) and the assumption about the mis-reported version $\tilde{P}_i$ from Definition 1. □

Corollary 3 implies that IC can be achieved easily if the ground truth distribution $P^*$ is (assumed to be) known. Compared to the case where $P_N$ is the reference, truthful reporting is the equilibrium (i.e., each vendor has the incentive to report truthfully) because the truthfully reported data collectively form $P_N$ and is used to evaluate each data vendor's data. In this case, the truthful reporting of each vendor affects each other through the formation of $P_N$. In contrast, if $P^*$ is known and used as the reference, each vendor still has the incentive to report truthfully because $P^*$ is used to evaluate their data; however, the truthful reporting of each vendor no longer affects other vendors.

### A.2.4 VALUATION ERROR FROM USING A REFERENCE

Sec. 5 describes a theoretically justified choice for a reference $P_{\text{ref}}$ to be used in place of $P^*$ in $\Upsilon$ in Eq. (1), and define an approximate $\hat{\Upsilon} := -d(P_{\text{ref}}, P)$. Then, the valuation error $|\Upsilon(P) - \hat{\Upsilon}(P)|$ from using this reference $P_{\text{ref}}$ should ideally be small. This error is directly upper bounded by the MMD between $P_{\text{ref}}$ and $P^*$, as follows.

**Lemma 5.** For a choice of reference $P_{\text{ref}}$, define $\hat{\Upsilon}(P) := -d(P_{\text{ref}}, P)$ as the approximate version of $\Upsilon(P)$ in Eq. (1). Then,

$$\forall P, |\Upsilon(P) - \hat{\Upsilon}(P)| \leq d(P_{\text{ref}}, P^*) .$$

*Proof.* Apply the triangle inequality of MMD to the definitions of $\Upsilon$ and $\hat{\Upsilon}$.

$$|\Upsilon(P) - \hat{\Upsilon}(P)| = |d(P, P^*) - d(P_{\text{ref}}, P^*)|$$
$$\leq d(P, P_{\text{ref}}) .$$

□

Lemma 5 implies that a better reference distribution (i.e., with a lower MMD to $P^*$) leads to a better approximate $\hat{\Upsilon}$ (i.e., with a lower error). Hence, we have specifically obtained theoretical results to upper bound the MMD between our considered choices for reference: $P_N$ via Lemma 6, which is be directly combined with Lemma 5 to derive the error guarantee, described next.

**Error from using $P_N$ as the reference.** We first provide an upper bound on $d(P_N, P^*)$ in Lemma 6, and then combine it with Lemma 5 into Proposition 2.

**Lemma 6.** The sampling distribution $P_N$ for the union $D_N$ satisfies $d(P_N, P^*) \leq \varepsilon_N d(Q_N, P^*)$ .

*Proof of Lemma 6.* The proof is a direct application of Observation 1 and Lemma 2. □

Lemma 6 provides an upper bound on the MMD between $P_N$ and $P^*$, which linearly depends on $\varepsilon_N$ and $Q_N$. A lower $\varepsilon_N$ or a lower $d(Q_N, P^*)$ leads to a smaller $d(P_N, P^*)$, making $D_N$ a better reference.

*Proof of Proposition 2.* It directly combines Lemma 6 and Lemma 5. □

**Using a general definition of reference $P_{\text{ref}}$.** One might ask: how to adapt and interpret the method and results w.r.t. a general definition of reference $P_{\text{ref}}$? This question can arise from practical settings where neither $P^*$ nor $P_N$ is usable: $P^*$ is unavailable in practice; some additional knowledge about $P_N$ might be available to inform that $P_N$ is too far (in the MMD sense) to $P^*$ to be a good reference (e.g., each local distribution $P_i$ is very statistically different from $P^*$, as mentioned in Sec. 5). In this scenario, it is of interest to consider the above question, so as to understand how to obtain or analyze an alternative reference.

Indeed, a similar analysis is enabled by Lemma 5, which is due to the analytic properties of Huber and MMD. In other words, for some generic reference $P_{\text{ref}}$, suppose it is known or can be derived that, $d(P_{\text{ref}}, P^*) \leq \varepsilon_{\text{ref}}$ (such as $P_N$ in Lemma 6), then the theoretical guarantee of comparing distributions using $P_{\text{ref}}$ (e.g., Proposition 1 for $P^*$, Theorem 1 for $P_N$) can be adapted by plugging this condition $d(P_{\text{ref}}, P^*) \leq \varepsilon_{\text{ref}}$. Furthermore, in a similar vein, the IC can be achieved, by plugging $d(P_{\text{ref}}, P^*) \leq \varepsilon_{\text{ref}}$, into Corollary 2, though the degree of IC (i.e., whether exact or $\gamma$-IC) depending on the tightness of the upper and lower bounds in Corollary 2 may vary.

As such, part of our theoretical analysis, as described above, is general and can be flexibly adapted to a general definition of the reference $P_{\text{ref}}$, through the knowledge of $d(P_{\text{ref}}, P^*) \leq \varepsilon_{\text{ref}}$. In this light, we highlight that our choice of $P_N$ as the reference, is amenable for theoretical analysis precisely because we can derive $d(P_{\text{ref}}, P^*) \leq \varepsilon_{\text{ref}}$ for when $P_{\text{ref}} = P_N$, by applying Observation 1. Consequently, this consideration w.r.t. a more general $P_{\text{ref}}$ naturally inspires future explorations: What are alternatives to $P_N$ as a reference? What is the "best" $P_{\text{ref}}$ when $P^*$ is unavailable, and what are the necessary assumptions? An choice adopted by some existing works (Tay et al., 2022; Wei et al., 2021) is an additional generative distribution learnt on $P_N$: Train a generative adversarial network (GAN) on the data $D_N \sim P_N$ and then use the generated data from this trained GAN as the reference (in place of $P^*$, as an alternative to $P_N$). The empirical results obtained by Tay et al. (2022); Wei et al. (2021) suggest that this is a promising approach, but unfortunately little theoretical analysis (such as Proposition 2) is provided. Thus, investigating the theoretical properties of this such a trained generative distribution forms an interesting future direction. Specifically, it is interesting to determine the generative distribution can provide the similar guarantee as $P_N$. A possible motivation for going beyond $P_N$ to adopt the generative distribution can be the privacy properties of a generative distribution (e.g., GAN) (Lin et al., 2021), which can be appealing in a data marketplace.

### A.3 Updated IC Definition and Results

==Updated theoretical results according to the updated definition of IC.==

For a vendor $i$, the true observed distribution is $P_i$ and the true observed sample $D_i \sim P_i$. The reported sample is $\tilde{D}_i$ which may or may not equal to $D_i$. Denote $\tilde{P}_i$ as the underlying sample distribution from which $\tilde{D}_i$ is sampled, namely $\tilde{D}_i \sim \tilde{P}_i$. In particular, for each data vendor $i$, the misreported $\tilde{P}_i \in \{Q; d(Q, P_i) \leq \kappa_i\}$ for some $\kappa_i \geq 0$ and denote $\kappa := \max_i \kappa_i$ . Intuitively, $\kappa_i$ is the (maximum) degree of misreporting for vendor $i$. The definition of $\tilde{P}_i$ is mainly for the clarity in theoretical discussion and the expression of $\tilde{P}_i$ may not be known or available. Denote $-i$ as a shorthand set notation for $N \setminus \{i\}$ .

**Definition 3** ($\gamma$-incentive compatibility)**.** The valuation function $\Upsilon$ is $\gamma$-incentive compatible if:

$$\Upsilon(P_i; \{P_{i'}\}_{i' \in -i}, \cdot) \geq \Upsilon(\tilde{P}_i; \{P_{i'}\}_{i' \in -i}, \cdot) - \gamma \,.$$

Here the value is used as the utility of vendor $i$, which is a more commonly used terminology in the literature of IC (Balseiro et al., 2022; Balcan et al., 2019).

Note that $\Upsilon()$ may have additional dependence such as $P^*$, which will be made explicit later where applicable.

In the ideal case, there is access to $P^*$, so the definitions of $\Upsilon, \nu$ have a dependence on them. However, in practice, due to no access to $P^*$, we define the approximate version $\hat{\Upsilon}(P_i; P_N) := -d(P_i, P_N)$ and correspondingly $\hat{\nu}(D_i; D_N) := -\hat{d}(D_i, D_N)$ . The notational dependence on $P_N$ and $D_N$ respectively is made explicit for clarity.

**Ideal case.**  In the ideal case where $P^*$ is available, the $\Upsilon$ as in Eq. (1) has a dependence on the ground truth $P^*$ and does *not* directly depend on $\{P_{i'}\}_{i' \in -i}$. In other words, in this optimal case, the IC is *not* conditioned on every other vendor reporting truthfully. This is because of the access to $P^*$. But the case for IC for the practical case is difference in that there is an explicit dependence on $\{P_{i'}\}_{i' \in -i}$, as described later.

**Proposition 4.**  $\Upsilon(P_i)$ from Eq. (1) is $\kappa$-IC.

*Proof of Proposition 4.*  Recall that $\Upsilon(P_i) = -d(P_i, P^*)$, then,

$$
\begin{aligned}
\Upsilon(\tilde{P}_i) &= -d(\tilde{P}_i, P^*) \\
&\geq -d(P_i, P^*) - d(P_i, \tilde{P}_i) \\
&= \Upsilon(P_i) - d(P_i, \tilde{P}_i) \\
&\geq \Upsilon(P_i) - \kappa_i
\end{aligned}
$$

where first inequality is via the triangle inequality of the MMD and the second inequality is by the definition of $\kappa_i$ .

Applying the definition of completes the proof, namely,

$$
\forall i \in N, \Upsilon(\tilde{P}_i) \geq \Upsilon(P_i) - \kappa_i \geq \Upsilon(P_i) - \kappa .
$$

$\square$

The relaxation $\kappa$ is the upper bound on the "degree" of misreporting over all vendors. Proposition 4 is the updated version of our result (Corollary 3) which was w.r.t. the previous definition of IC.

**Practical case.**  In the practical case, namely $P^*$ is not available, we propose to use $P_N$ in its stead as in Eq. (3) (recalled here for easy reference):

$$
\hat{\Upsilon}(P_i; P_N) := -d(P_i, P_N) .
$$

Recall that the approximation error of $\hat{\Upsilon}$ to $\Upsilon$ is bounded as in Proposition 2 and we focus on deriving the IC guarantee for $\hat{\Upsilon}$ .

Note that $P_N$ depends on $P_i$ as in Observation 1, so if vendor $i$ indeed misreports (i.e., $\tilde{P}_i \neq P_i$), we use $\tilde{P}_{N,i}$ to denote the corresponding aggregate distribution. In other words $P_N$ is the sampling distribution for $D_N := \cup_{i \in N} D_i$ (i.e., $D_N \sim P_N$) while $\tilde{P}_{N,i}$ is the sampling distribution of $\tilde{D}_{N,i} := \tilde{D}_i \cup D_{-i}$ (i.e., $\tilde{D}_{N,i} \sim \tilde{P}_{N,i}$).

**Proposition 5.**  $\hat{\Upsilon}$ is $\gamma_{\hat{\Upsilon}}$-IC where $\gamma_{\hat{\Upsilon}} := \max_i \kappa_i \frac{m_{-i}}{m_N}$ .

*Proof of Proposition 5.*  From the definition of $\hat{\Upsilon}$:

$$
\begin{aligned}
\hat{\Upsilon}(\tilde{P}_i, \tilde{P}_{N,i}) &= -d(\tilde{P}_i, \tilde{P}_{N,i}) \\
&= -\frac{m_{-i}}{m_N} d(\tilde{P}_i, P_{-i}) \\
&\geq -\kappa_i \frac{m_{-i}}{m_N} + \hat{\Upsilon}(P_i, P_N) .
\end{aligned}
\tag{4}
$$

The last inequality (i.e., Eq. (4)) is derived from (by multiplying $-1$ to) the following derived inequality: Note that

$$
\begin{aligned}
d(\tilde{P}_i, P_{-i}) &\leq d(P_i, \tilde{P}_i) + d(P_i, P_{-i}) \\
&\leq \kappa_i + \frac{m_N}{m_{-i}} d(P_i, P_N) \\
&\leq \kappa_i - \frac{m_N}{m_{-i}} \hat{\Upsilon}(P_i, P_N)
\end{aligned}
$$

where the first inequality applies the triangle inequality of the MMD and the second inequality applies the definition of $\kappa_i$ and also Lemma 4. We highlight that Lemma 4 is uniquely available to the combined design choices of the Huber model and the MMD.

The proof is complete by applying the definition of $\gamma_{\hat{\Upsilon}}$ to Eq. (4):

$$
\hat{\Upsilon}(\tilde{P}_i, \tilde{P}_{N,i}) \geq -\kappa_i \frac{m_{-i}}{m_N} + \hat{\Upsilon}(P_i, P_N) \geq \hat{\Upsilon}(P_i, P_N) - \underbrace{\max_i \kappa_i \frac{m_{-i}}{m_N}}_{\gamma_{\hat{\Upsilon}}}.
$$

$\square$

The relaxation $\gamma_{\hat{\Upsilon}}$ depends on both the degree of misreporting of the vendors and the relative size (i.e., $m_i/m_N$) of the vendors. In particular, if $m_i = m_{i'}, \forall i \neq i'$, then $\hat{\Upsilon}$ is $(\kappa \frac{n-1}{n})$-IC. While this may seem to be a better IC guarantee than in Proposition 4 since $\kappa \frac{n-1}{n} < \kappa$, importantly, $\hat{\Upsilon}$ has an (approximation) error as in Lemma 6. In other words, while in some cases $\hat{\Upsilon}$ provides a better IC guarantee than $\Upsilon$, the cost is that its provided value is not accurate as $\Upsilon$, and this is due to the practical constraint that $P^*$ is *not* available.

Proposition 5 is the updated version of our results (Corollary 1, Proposition 3) which was w.r.t. the previous definition of IC.

# B    ADDITIONAL DISCUSSION ON RELATED WORKS.

In addition to the existing works mentioned in Sec. 2, Bian et al. (2021); Jia et al. (2018; 2019); Yoon et al. (2020) also share a similar dependence on a given reference dataset. Consequently, it is unclear how to apply these methods *without* a given reference dataset.

Next, we elaborate the works that relax the assumption of a given reference and better contrast their differences from our work, specifically in how they relax this assumption.

Chen et al. (2020, Assumption 3.2) require that $P^*$ must follow a known parametric form and the posterior of the parameters is known. Chen et al. (2020) assumes that each data vendor collects data from the ground truth data distribution (parametrized by some unknown parameters) and performs the analysis on whether a data vendor $i$ will report untruthfully when *everyone* else is reporting truthfully. Specifically, the reference is the aggregate of all vendors excluding the vendor $i$ itself. It is unclear how heterogeneity can be formalized in their theoretical analysis, which assumes the vendors are collecting data from the same (ground truth) data distribution. (Tay et al., 2022) directly assumes that the aggregate dataset $D_N$ or the aggregate distribution $P_N$ is a sufficiently good representation of $P^*$ and thus provides a good reference, *without* formally justifying it or accounting for the cause or effect of heterogeneity. Wei et al. (2021, Assumption 3.1) require that for any $i, i'$, the densities of $P_i$ and $P_{i'}$ must lie on the same support, which is difficult to guarantee because $Q_i, Q_{i'}$ can have different supports (resulting in $P_i, P_{i'}$ having different supports). (Wei et al., 2021) uses the $f$-divergence specifically the KL divergence, which requires an assumption that the densities of the data distributions (of all the vendors) to lie on the same support (Wei et al., 2021, Assumption 3.1). This assumption can be difficult to satisfy in practice since the analytic expression of the density of the data distribution is often complex and unknown. To illustrate, our experiments consider heterogeneity in the form of mixture between MNIST and FaMNIST; it is unclear how to satisfy the assumption that the densities of the distributions of MNIST and FAMNIST lie on the same support. Moreover, their method does not formally model heterogeneity or its effect on the value of data.

Furthermore, a similarity in these works Chen et al. (2020); Tay et al. (2022); Wei et al. (2021) is how they leverage the "majority" to construct a reference either implicitly or explicitly. Specifically, in (Chen et al., 2020), the reference for vendor $i$ is the aggregate of every vendor's dataset except $i$; in (Tay et al., 2022; Wei et al., 2021) the reference is constructed by utilizing both the aggregate dataset $D_N$ and additionally some synthetically generated data (Tay et al. (2022) uses the MMD-GAN while Wei et al. (2021) uses the $f$-divergence GAN). We highlight that these works did not provide a theoretical analysis on "how good" their respective reference is, namely, what is the error from using the reference instead of using the ground truth? In contrast, we answer this question via Proposition 2 and derive IC guarantees that show a clear dependence on the quality of the "majority" (e.g., $d(P_{-i}, P^*)$ in Corollary 1).

## B.1 COMPARISON WITH ALTERNATIVE DISTANCES/DIVERGENCES

We provide some additional comparative discussion with alternative distances/divergences (to our adopted MMD) that have already been adopted for the purpose of data valuation.

**Comparison between Ours and MMD$^2$.** Empirically, Ours (i.e., MMD-based) and MMD$^2$ (Tay et al., 2022) perform similarly across the investigated settings (including empirical convergence, ranking data distributions and incentive compatibility), which is unsurprising since the numerical values only differ by a square. Although MMD$^2$ has an unbiased estimator (Gretton et al., 2012, Eq. (4)) while MMD, to our knowledge, only has a biased estimator (Gretton et al., 2012, Eq. (6)), this advantage does not seem significant, since the convergence results in App. C.3.1 demonstrate similar convergences for Ours and MMD$^2$. Recall that Sec. 6, we highlighted that the implemented estimator for MMD$^2$ is *not* obtained from taking the square of that for MMD. Nevertheless, we have also tried this implementation of directly squaring the estimator for MMD to be the estimator of MMD$^2$ but did not observe a significant difference in the empirical results.

On the other hand, from a theoretical perspective, the difference in terms of the implications, is more significant. This is primarily due to the analytic properties of MMD, which are *not* also satisfied by MMD$^2$, such as the triangle inequality, used to derive Corollary 1, and the property with Huber model (i.e., Eq. (2)) which is used to derive Lemma 6 and Theorem 1. It is an interesting in the future to explore similar results for MMD$^2$.

**Comparison between Ours and the Wasserstein metric.** Similar to MMD, the Wasserstein metric, also known as the optimal transport (OT) distance (Kantorovich, 2006) also satisfies the axioms of a metric, in particular the triangle inequality. This can make the Wasserstein metric a promising choice for our setting. However, MMD seems to have two important advantages —one theoretical and the other practical—: (i) Under the Huber model, MMD enables a simple and direct relationship (i.e., Eq. (2)) that precisely characterizes how the heterogeneity (formalized by Huber) of a distribution affects its value. It is unclear how or whether the Wasserstein metric can provide the same relationship. (ii) The definition of the Wasserstein metric (involving taking an infimum over the couplings of distributions) makes its value difficult to obtain (compute or approximate) in practice. Indeed, LAVA's official implementation (Just et al., 2023, Github repo) does not directly compute/approximate the 2-Wasserstein distance, but instead obtains the calibrated gradients as a surrogate (Just et al., 2023, Sec. 3.2), which are not explicitly guaranteed to approximate the 2-Wasserstein-based values. In contrast, our method directly approximates the MMD (with the estimator (Gretton et al., 2012, Eq. (6))) with a clear theoretical guarantee (Lemma 1). In other words, though the Wasserstein metric satisfies the appealing theoretical properties, in implementation, a valuation based on the Wasserstein metric is difficult to obtain directly, and instead some surrogate is obtained. Moreover, it has not yet been guaranteed that this surrogate also satisfies (possibly approximately) the same appealing theoretical properties that might make the Wasserstein metric a promising choice.

**Comparison between Ours and $f$-divergences.** The $f$-divergences family presents a rich choice (since it contains many specific divergences such as Kullback-Leibler, total variation and etc.) and is also adopted in existing works such as (Chen et al., 2020, Definition 4.5), (Wei et al., 2021, Algorithm 1) and (Agussurja et al., 2022, Eq.(1)) for the theoretical properties of the adopted $f$-divergence (or variant). For instance, the Kullback-Leibler (KL) divergence satisfies the required (Wei et al., 2021, Assumption 3.4) for the proposed method, while the extended KL divergence proposed by Agussurja et al. (2022, Sec. 3) enables a decomposition of the terms to simplify the analysis.

These theoretical properties notwithstanding, our adopted MMD has a clear advantage over the $f$-divergence with important practical implications. A commonly made assumption with using the $f$-divergence $f(P\|Q)$ is the absolute continuity of $P$ w.r.t. $Q$, since otherwise the division-by-zero makes the definition ill-behaved. This assumption is difficult to satisfy or even verify in practice, especially for complex and high-dimensional data distributions. In contrast, the sample-based definition of MMD does *not* require such an assumption, making its application to complex and high-dimensional data distributions easier (in the sense that the user does not have to worry about a difficult-to-satisfy assumption). Another important practical implication due to the difference in the definitions of $f$-divergences and MMD is that: it is more direct and easier to approximate MMD (e.g., using a sample-based approach (Gretton et al., 2012, Eq. (6)) and with a theoretical guarantee as in Lemma 1). In contrast, the definition of the $f$-divergence $f(P\|Q)$ that directly depends on the density functions of $P, Q$ adds to the difficulties of estimating it in practice, as it requires estimating the density functions (or at least the ratio of the density functions). This is difficult for complex and high-dimensional data distributions and may require simplifying assumptions of the parametric form of the distributions (e.g., $P, Q$ are both multi-variate Gaussian (Agussurja et al., 2022) to enable a closed-form expression for KL).

## C  ADDITIONAL EXPERIMENTAL SETTINGS AND RESULTS

### C.1  DATASET LICENSES AND COMPUTATIONAL RESOURCES

MNIST (LeCun et al., 1990): Creative Commons Attribution-Share Alike 3.0. EMNIST (Cohen et al., 2017): CC0: Public Domain. FaMNIST (Xiao et al., 2017): The MIT License (MIT). CIFAR-10 and CIFAR-100 (Krizhevsky, 2009): The MIT License (MIT). CaliH (Kelley Pace & Barry, 1997): CC0: Public Domain. KingH (Harlfoxem, 2016): CC0: Public Domain. TON (Moustafa, 2021): CC0: Public Domain. UGR16 (Maciá-Fernández et al., 2018): CC0: Public Domain. Credit7 (Narayanan, 2022): CC0: Public Domain. Credit31 (Andrea Dal Pozzolo & Bontempi, 2015): CC0: Public Domain. Census15, Census17 (Muonneutrino, 2019): CC0: Public Domain.

Our experiments are run on a server with Intel(R) Xeon(R) Gold 6226R CPU @2.90GHz and 4 NVIDIA GeForce RTX 3080's (each with 10 GBs memory). We run our experiments for 5 independent trials to report the average and standard errors.

### C.2  ADDITIONAL EXPERIMENTAL SETTINGS

Table 5 provides an overall summary of the experimental settings.

Table 5: Datasets, used ML models $\mathbf{M}$ and $n, m_i, \varepsilon_i$.

| Setting | $P^*$ | $Q$ | $\mathbf{M}$ | n | $m_i$ | $\varepsilon_i$ |
|---|---|---|---|---|---|---|
| Class. | MNIST | EMNIST | CNN | 5 | 10000 | $(i-1)/n$ |
| | MNIST | FaMNIST | CNN | 10 | 10000 | $(i-1)/n$ |
| | CIFAR10 | CIFAR100 | ResNet-18 | 5 | 10000 | $(i-1)/n$ |
| | Credit7 | Credit31 | LogReg | 5 | 5000 | $(i-1)/(4n)$ |
| | TON | UGR16 | LogReg | 5 | 4000 | $(i-1)/(4n)$ |
| Regress. | CaliH | KingH | LR | 10 | 2000 | $(i-1)/n$ |
| | Census15 | Census17 | LR | 5 | 4000 | $(i-1)/n$ |

**Additional dataset preprocessing details.**  For CaliH and KingH, the pre-processing is passing the original (raw) data through a neural network (i.e., feature extractor) with the last layer having 10 units (Agussurja et al., 2022; Xu et al., 2021b). Hence, the dimensionality after preprocessing is 10.

Credit7 has 7 features while Credit31 has 31 features, so the top 7 principal components are kept for Credit31. TON and UGR16 have the same feature space including features such as packet byte, source and destination IP addresses, but non-numerical features such as IP addresses are converted to one-hot encoding.

For TON and UGR16, which share the same feature space containing features such as source and destination IP addresses, packet size, network protocol and etc, we adopt the one-hot encoding for non-numerical features (i.e., source and destination IPs, network protocol) and perform standard scaling of the numerical features (i.e., packet size). The dimensionality after preprocessing is 22.

**ML model M specification details.** For MNIST vs. EMNIST and MNIST vs. FaMNIST, a standard 2-layer convoluntional neural network (CNN) is adopted. For CIFAR10 vs. CIFAR100, the ResNet-18 (He et al., 2016) is adopted. For both Credit7 vs. Credit31, and TON vs. UGR16, a standard logistic regression with the corresponding input sizes is adopted. The specific implementation (i.e., source code) is provided in the supplementary material for reference.

**Additional implementation details on Ours cond.** Different from directly computing MMD between the features, this extension of our method aims to additionally utilize the label information contained within each $D_i$. Recall that each single data point in $D_i$ is paired feature-label, so this extension aims to exploit such information. Theoretically, MMD is well-defined w.r.t. distributions, be it distributions over only the features (i.e., $P_X$) or conditional distributions of the labels given features (i.e., $P_{Y|X}$). For implementation, we need a representation for $P_{Y|X}$ for a $D_i$. In our implementation, we train a machine learner $\mathbf{M}_i := \mathbf{M}(D_i)$ on $D_i$ and use it to construct the empirical representation for $P_{Y|X}$. Given a reference $D_{\text{val}}$, we collect the set of predictions (denoted as $\mathbf{M}_i(D_{\text{val}})$) of $\mathbf{M}_i$ on this reference, as the empirical representation of $P_{Y|X}$. Subsequently, we compute the data values, namely negated MMD between $\mathbf{M}_i(D_{\text{val}})$ and, the labels of $D_{\text{val}}$ (i.e., the left column of Tables 1 and 2). Note that the predictions (i.e., $\mathbf{M}_i(D_{\text{val}})$) are probability vectors in the $C-1$-probability simplex $\Delta(C)$ for classification with $C$ classes, or real-values for regression. For MMD computation, a one-hot encoding of the labels in $D_{\text{val}}$ for classification is performed; no additional processing is required for the labels for regression.

## C.3 ADDITIONAL EXPERIMENTAL RESULTS

### C.3.1 EMPIRICAL CONVERGENCE TO INVESTIGATE SAMPLE EFFICIENCY

A desirable solution to the problem statement (in Sec. 3) is a sample-efficient actionable policy that can correctly compare $\Upsilon(P)$ vs. $\Upsilon(P')$ from comparing $\nu(D)$ vs. $\nu(D')$, even if the sizes of $D, D'$ are relatively small. This is because in practice, there is no direct access to $P, P'$ but only $D, D'$; moreover, the sizes of $D, D'$ may not be very large. In this regard, we evaluate $\{\nu(D_i)\}_{i \in [n]}$ against $\{\Upsilon(P_i)\}_{i \in [n]}$ w.r.t. varying sizes $m_i$, since as $m_i$ becomes very large $\nu(D_i)$ converges to $\Upsilon(P_i)$ (Lemma 1).

**Setting.** We implement $\Upsilon(P_i)$ as approximated by a $\nu(D_i^*)$ where $D_i^* \sim P_i$ with a large size $m_i^*$ (e.g., 10000 for $P^* = $ MNIST vs. $Q = $ EMNIST). Denote the values of the samples as $\boldsymbol{\nu}_{m_i} := \{\nu(D_i)\}_{i \in [n]}$ where the sample size $m_i = |D_i|$ and the approximated ground truths as $\boldsymbol{\nu}^* := \{\nu(D_i^*)\}_{i \in [n]}$; in this way, $\boldsymbol{\nu}^*$ is well-defined respectively for each comparison baseline (i.e., *not* our MMD definition Eq. (1)). We highlight that each $\nu$ (i.e., each baseline) is evaluated against its corresponding $\boldsymbol{\nu}^*$ to demonstrate the empirical convergence. This is to examine the practical applicability of each $\nu$ when the sizes of the provided $\{D_i\}_{i \in N}$ are limited.

We consider three criteria—the $\ell_2$ and $\ell_\infty$ errors and the number of pair-wise inversions as follows, $\|\boldsymbol{\nu}_{m_i} - \boldsymbol{\nu}^*\|_2, \|\boldsymbol{\nu}_{m_i} - \boldsymbol{\nu}^*\|_\infty$ and $\texttt{inversions}(\boldsymbol{\nu}_{m_i}, \boldsymbol{\nu}^*) := (1/2) \sum_{i,i' \in [n], i \neq i'} \mathbb{1}(\nu(D_i^*) > \nu(D_{i'}^*) \wedge \nu(D_i) < \nu(D_{i'}))$. In other words, if the conclusion via $\nu(D_i)$ vs. $\nu(D_{i'})$ *differs* from that via $\nu(D_i^*)$ vs. $\nu(D_{i'}^*)$, it is an inversion. The datasets and the comparison baselines follow Sec. 6.

**Results.** Plotting these criteria against the sample size $m_i$ compares the sample efficiency of the baselines in how quickly the sample values converge to the ground truth values, demonstrating the practical applicability of the methods. Specifically, Figs. 3 to 7 show that our method utilizing the MMD in general converges the most quickly and hence is the most sample-efficient overall.[8]

---

[8]Note that on three datasets (i.e., CIFAR10, TON and Credit), the baseline LAVA did not complete due to a runtime error (see Github issue) of a required package OTDD.

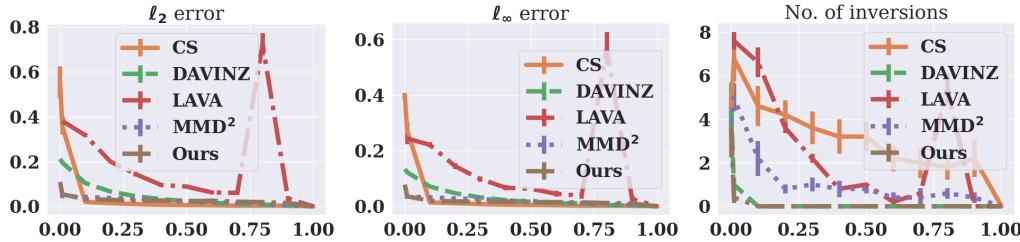

Figure 3: From left to right: $\ell_2$ error, $\ell_\infty$ error and number of inversions. $x$-axis shows the ratio $m_i/m_i^*$ between sample size and the ground truth sample size. Averaged over 5 independent trials and the error bars reflect the standard errors. $P^*$ = MNIST vs. $Q$ = EMNIST. $n = 5, m_i^* = 10000$.

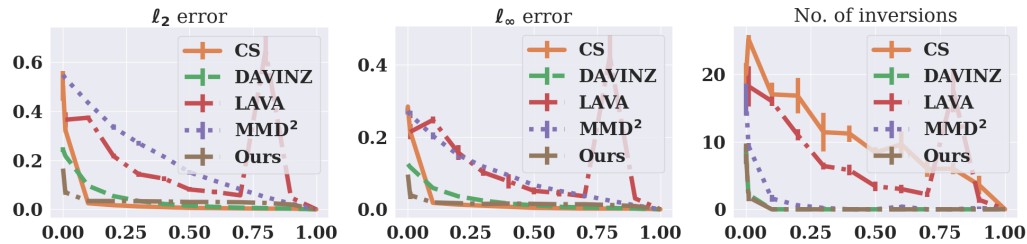

Figure 4: From left to right: $\ell_2$ error, $\ell_\infty$ error and number of inversions. $x$-axis shows the ratio $m_i/m_i^*$ between sample size and the ground truth sample size. Averaged over 5 independent trials and the error bars reflect the standard errors. $P^*$ = MNIST vs. $Q$ = FaMNIST. $n = 10, m_i^* = 10000$.

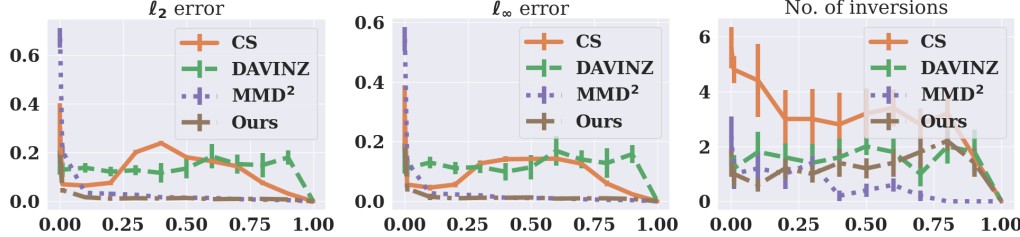

Figure 5: From left to right: $\ell_2$ error, $\ell_\infty$ error and number of inversions. $x$-axis shows the ratio $m_i/m_i^*$ between sample size and the ground truth sample size. Averaged over 5 independent trials and the error bars reflect the standard errors. $P^*$ = TON vs. $Q$ = UGR16. $n = 5, m_i^* = 10000$.

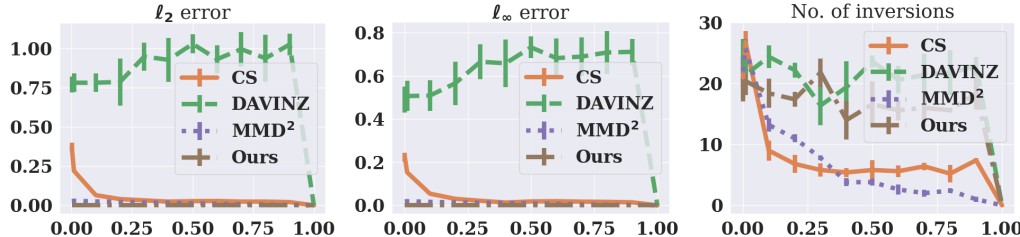

Figure 6: From left to right: $\ell_2$ error, $\ell_\infty$ error and number of inversions. $x$-axis shows the ratio $m_i/m_i^*$ between sample size and the ground truth sample size. Averaged over 5 independent trials and the error bars reflect the standard errors. $P^*$ = CIFAR10 vs. $Q$ = CIFAR100. $n = 10, m_i^* = 10000$.

### C.3.2 ADDITIONAL RESULTS FOR HUBER SETTING

We present the remaining results for the Huber setting for Credit7/Credit31, MNIST/EMNIST, and MNIST/FaMNIST in Tables 6 to 8, respectively. Note that the results for pearson$(\nu, \zeta)$ are obtained

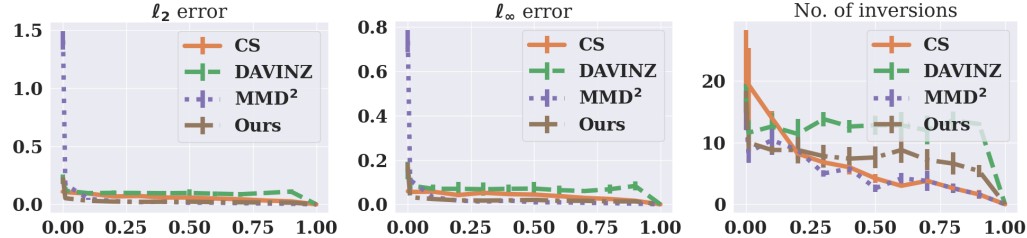

Figure 7: From left to right: $\ell_2$ error, $\ell_\infty$ error and number of inversions. $x$-axis shows the ratio $m_i/m_i^*$ between sample size and the ground truth sample size. Averaged over 5 independent trials and the error bars reflect the standard errors. $P^* =$ Credit7 vs. $Q =$ Credit31. $n = 10, m_i^* = 10000$.

w.r.t. an available $D_{\text{val}} \sim P^*$ as the reference. Hence, our method directly uses $D_{\text{val}}$ (since it is equivalent to $D^*$ by definition) in Eq. (1) when $D_{\text{val}}$ is available. Our method uses $D_N$ (i.e., Eq. (3)) when $D_{\text{val}}$ is unavailable (i.e., the right columns). We highlight that Ours cond. is not applicable for when $D_{\text{val}}$ is unavailable because the conditional distribution required is not well-defined when the reference (i.e., $D_N$) is from a Huber distribution.

Table 6: **Classification**: $P^* =$ Credit7, $Q =$ Credit31.

Table 7: **Classification**: $P^* =$ MNIST, $Q =$ EMNIST.

| Baselines | pearson($\nu, \zeta$) | pearson($\hat{\nu}, \zeta$) |
|---|---|---|
| LAVA | 0.414(0.15) | 0.079(0.31) |
| DAVINZ | **0.878(0.06)** | -0.099(0.15) |
| CS | -0.813(0.03) | -0.101(0.13) |
| MMD$^2$ | 0.849(0.03) | 0.561(0.06) |
| Ours | 0.848(0.03) | **0.604(0.31)** |
| Ours cond. | 0.762(0.04) | N.A. |

| Baselines | pearson($\nu, \zeta$) | pearson($\hat{\nu}, \zeta$) |
|---|---|---|
| LAVA | -0.543(0.07) | 0.685(0.03) |
| DAVINZ | **0.977(0.00)** | 0.105(0.04) |
| CS | 0.760(0.08) | -0.984(0.00) |
| MMD$^2$ | 0.931(0.01) | 0.970(0.01) |
| Ours | 0.950(0.01) | **0.984 (0.01)** |
| Ours cond. | 0.971(0.01) | N.A. |

Table 8: **Classification**: $P^* =$ MNIST, $Q =$ FaMNIST.

| Baselines | pearson($\nu, \zeta$) | pearson($\hat{\nu}, \zeta$) |
|---|---|---|
| LAVA | -0.810(0.06) | 0.244(0.09) |
| DAVINZ | **0.864(0.05)** | 0.113(0.13) |
| CS | 0.314(0.13) | -0.810(0.03) |
| MMD$^2$ | 0.750(0.05) | **0.740(0.05)** |
| Ours | 0.747(0.05) | 0.739(0.05) |
| Ours cond. | 0.825(0.07) | N.A. |

### C.3.3 RESULTS FOR NON-HUBER SETTING

We investigate two non-Huber settings: (1) additive Gaussian noise; (2) different supports (Xu et al., 2021a), which is also generalized to an interpolated setting where the interpolation is between the different supports.

**Setting.** (1) Additive Gaussian noise: Among $n = 10$ vendors, the dataset $D_i \sim P_i \coloneqq$ MNIST $+ \mathcal{N}(\mathbf{0}, \varepsilon_i \times \mathbf{I})$ where $|D_i| = m_i = 5000$ and the $\varepsilon_i$'s are $[0, 0.02, \ldots, 0.18]$. Note that though $P^* =$ MNIST, each $P_i$ is *not* Huber. (2) The supports of $P^*$ that each vendor can sample from are different: on MNIST, vendor 1 only collects images of digits 0, while vendor 10 collects images of all 10 digits (called classimbalance (Xu et al., 2021a)). Intuitively, vendor 10 has access to $P^*$ so its data should be the most valuable. **Results.** Tables 9 and 10 show that, when $D_{\text{val}} \sim P^*$ is available, the methods (i.e., DAVINZ, CS) that can effectively utilize $D_{\text{val}}$ can outperform our method. However, without $D_{\text{val}}$, these methods still underperform our method, especially under

additive Gaussian noise. We believe it could be because these methods were not specifically designed to account for heterogeneity (which could be caused by noise), since CS performs comparably well under the class imbalance setting where the heterogeneity is due to the supports of the vendors being different instead of random noise. In particular, we find that all baselines perform sub-optimally under the additive Gaussian noise setting and when there is no clean $D_{\text{val}}$ available, which can be an interesting future direction. A possible reason is that the Gaussian noise is completely uncorrelated to the features and "destroys" the information in the data, rendering the valuation methods ineffective.

Table 9: **Non-Huber**: additive Gaussian noise.

| Baselines | pearson$(\boldsymbol{\nu}, \boldsymbol{\zeta})$ | pearson$(\hat{\boldsymbol{\nu}}, \boldsymbol{\zeta})$ |
|---|---|---|
| LAVA | -0.255(0.17) | -0.037(0.20) |
| DAVINZ | 0.848(0.02) | -0.410(0.03) |
| CS | 0.902(0.03) | -0.934(0.01) |
| MMD$^2$ | -0.085(0.04) | -0.496(0.03) |
| Ours | **0.964(0.01)** | -0.169(0.06) |
| Ours cond. | 0.892(0.04) | -0.668(0.06) |

Table 10: **Non-Huber**: classimbalance.

| Baselines | pearson$(\boldsymbol{\nu}, \boldsymbol{\zeta})$ | pearson$(\hat{\boldsymbol{\nu}}, \boldsymbol{\zeta})$ |
|---|---|---|
| LAVA | 0.439(0.26) | 0.340(0.34) |
| DAVINZ | 0.807(0.00) | 0.081(0.00) |
| CS | 0.985(0.00) | 0.871(0.01) |
| MMD$^2$ | 0.780(0.00) | -0.894(0.00) |
| Ours | 0.923(0.00) | 0.557(0.01) |
| Ours cond. | **0.989(0.00)** | **0.911**(0.00) |

**Interpolated classimbalance setting.** In addition to the "discrete" class imbalance setting, we also investigate an interpolated setting as follows: For MNIST, $n = 5, m_i = 5000$, half of $D_i$ consists of images of first $2i + 1$ digits while the other half consists of images of all 10 digits. E.g., 2500 of $D_3$ are images of digits of $0 - 6$ while the other 2500 are images of digits of $0 - 9$. Effectively, each $D_i$ contains images of *all* 10 digits, but in different proportions which increase as $i$ increases from 1 to 5. For CIFAR-10, $n = 5, m_i = 10000$ with the same interpolation implementation. Results are in Table 11. Note that for the non-Huber setting here, since the heterogeneity is only in the supports of the data (i.e., features) and not the labels, the conditional distribution for Ours cond. is indeed well-defined and thus Ours cond. is applicable here. This is different from the Huber setting examined in Sec. 6.

Table 11: Interpolated class-imbalance setting on MNIST (left) and CIFAR-10 (right).

| Baselines | pearson$(\boldsymbol{\nu}, \boldsymbol{\zeta})$ | pearson$(\hat{\boldsymbol{\nu}}, \boldsymbol{\zeta})$ |
|---|---|---|
| LAVA | 0.459(0.24) | 0.195(0.26) |
| DAVINZ | 0.962(0.01) | 0.706(0.04) |
| CS | **0.977(0.00)** | **0.952(0.02)** |
| MMD$^2$ | 0.839(0.05) | -0.969(0.01) |
| Ours | 0.939 (0.02) | 0.770(0.04) |
| Ours cond. | 0.859(0.03) | 0.857(0.05) |

| Baselines | pearson$(\boldsymbol{\nu}, \boldsymbol{\zeta})$ | pearson$(\hat{\boldsymbol{\nu}}, \boldsymbol{\zeta})$ |
|---|---|---|
| LAVA | 0.790(0.06) | 0.679(0.09) |
| DAVINZ | 0.498(0.25) | 0.495(0.28) |
| CS | **0.974(0.01)** | **0.983(0.01)** |
| MMD$^2$ | 0.472(0.25) | 0.285(0.03) |
| Ours | 0.931(0.11) | 0.332(0.02) |
| Ours cond. | 0.846(0.04) | 0.905(0.04) |

### C.3.4 ADDITIONAL EXPERIMENTAL RESULTS FOR INCENTIVE COMPATIBILITY

Additional results for incentive compatibility under varying settings of $P, Q$ and $n$ are presented, in Figs. 8 and 9 and Tables 12 to 15.

Figs. 8 and 9 verify the observation that the mis-reporting vendor $i'$ has a negative change (i.e., decrease) in value. Moreover, the magnitude of the decrease in value is more significant for Ours than that for MMD$^2$. The additional set of quantitative result (i.e., the Pearson correlation coefficient between the GT data values and Ours or MMD$^2$) also confirms this observation. To elaborate, take the value corresponding to $i' = 2$ and MMD$^2$ in Table 12 as an example. The Pearson coefficient (i.e., 0.999) is between the GT data values (i.e., a vector of length $n = 5$) and the MMD$^2$ data values (i.e., a vector of length $n = 5$), under the setting that $i' = 2$ is mis-reporting. The results in Tables 12 to 15 show that the data values of both Ours and MMD$^2$ are very consistent with the GT values, importantly *without* having $D^* \sim P^*$ as the reference, thus verifying IC. However, a caveat is that both methods are not very effective in achieving IC when $i' = 1$. This is precisely because $D_N \sim P_N$ is used as the reference in place of $D^* \sim P^*$ as for GT, and expected from the theoretical results (i.e., Corollary 1 and Proposition 3). Specifically, IC can be achieved when $d(P_{-i}, P^*)$ is small (see

the if-condition in Corollary 1). When $i' = 1$, this is not satisfied, in other words $d(P_{-i}, P^*)$ would be large. This is because, $i' = 1$ is the "best" vendor in our experiment settings in that $P_{i'=1} = P^*$ (i.e., $\varepsilon_{i'=1} = 0$). Hence, if $i' = 1$ mis-reports, the remaining vendors in $N \setminus \{i'\}$ are unable to catch $i' = 1$ because the aggregate distribution $P_{-i'} = P_{-1}$ is not a very good reference (as compared to $P_{-i'}$ for $i' \neq 1$ is mis-reporting). Intuitively, if $i = 1$ is *not* mis-reporting (i.e., $i' \neq 1$), then $P_{-i'}$ contains the data from $P_1 = P^*$, and is thus a good reference.

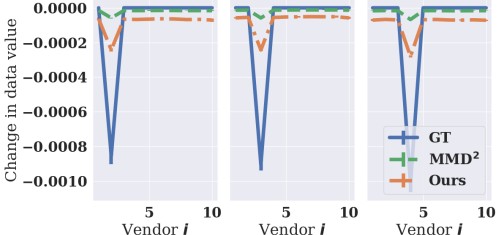

Figure 8: Change in data values for $P_{\mathrm{MNIST}}$ and $Q_{\mathrm{EMNIST}}$ with $n = 10$.

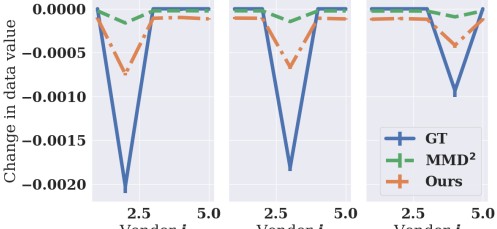

Figure 9: Change in data values for $P_{\mathrm{MNIST}}$ and $Q_{\mathrm{FaMNIST}}$ with $n = 5$.

Table 12: Average and standard error (over 5 independent trials) of Pearson coefficients with GT for $P_{\mathrm{MNIST}}$ and $Q_{\mathrm{EMNIST}}$ with $n = 5$, rows $i' = 2, 3, 4$ corresponding to Fig. 1.

| $i'$ | MMD$^2$ | Ours |
|---|---|---|
| 1 | 0.485 (0.03) | 0.476 (0.03) |
| 2 | 0.999 (0.00) | 0.999 (0.00) |
| 3 | 1.000 (0.00) | 1.000 (0.00) |
| 4 | 1.000 (0.00) | 1.000 (0.00) |
| 5 | 1.000 (0.00) | 1.000 (0.00) |

Table 13: Average and standard error (over 5 independent trials) of Pearson coefficients with GT for $P_{\mathrm{MNIST}}$ and $Q_{\mathrm{FaMNIST}}$ with $n = 5$, rows $i' = 2, 3, 4$ corresponding to Fig. 9.

| $i'$ | MMD$^2$ | Ours |
|---|---|---|
| 0 | 0.515 (0.01) | 0.506 (0.01) |
| 1 | 1.000 (0.00) | 1.000 (0.00) |
| 2 | 0.999 (0.00) | 0.999 (0.00) |
| 3 | 0.999 (0.00) | 0.999 (0.00) |
| 4 | 0.997 (0.00) | 0.998 (0.00) |

Table 14: Average and standard error (over 5 independent trials) of Pearson coefficients with GT for $P_{\mathrm{MNIST}}$ and $Q_{\mathrm{EMNIST}}$ with $n = 10$, rows $i' = 2, 3, 4$ corresponding to Fig. 8.

| $i'$ | MMD$^2$ | Ours |
|---|---|---|
| 0 | 0.386 (0.01) | 0.383 (0.01) |
| 1 | 0.997 (0.00) | 0.997 (0.00) |
| 2 | 0.997 (0.00) | 0.997 (0.00) |
| 3 | 0.999 (0.00) | 0.999 (0.00) |
| 4 | 0.998 (0.00) | 0.998 (0.00) |
| 5 | 0.998 (0.00) | 0.998 (0.00) |
| 6 | 0.999 (0.00) | 0.999 (0.00) |
| 7 | 0.999 (0.00) | 0.999 (0.00) |
| 8 | 0.998 (0.00) | 0.998 (0.00) |
| 9 | 0.993 (0.00) | 0.994 (0.00) |

Table 15: Average and standard error (over 5 independent trials) of Pearson coefficients with GT for $P_{\mathrm{MNIST}}$ and $Q_{\mathrm{FaMNIST}}$ with $n = 10$, rows $i' = 2, 3, 4$ corresponding to Fig. 2.

| $i'$ | MMD$^2$ | Ours |
|---|---|---|
| 1 | 0.364 (0.01) | 0.362 (0.01) |
| 2 | 0.999 (0.00) | 0.999 (0.00) |
| 3 | 0.997 (0.00) | 0.997 (0.00) |
| 4 | 0.999 (0.00) | 0.998 (0.00) |
| 5 | 0.998 (0.00) | 0.998 (0.00) |
| 6 | 0.995 (0.00) | 0.995 (0.00) |
| 7 | 0.996 (0.00) | 0.995 (0.00) |
| 8 | 0.954 (0.01) | 0.950 (0.01) |
| 9 | 0.852 (0.06) | 0.858 (0.05) |
| 10 | 0.997 (0.00) | 0.998 (0.00) |

## C.4 OBSERVED LINEAR SCALING W.R.T. $n$ AND $m$

We demonstrate the scalability of our method w.r.t. the number $n$ of vendors and the sample size $m$, in terms of execution time and memory (RAM and GPU).

**Plots showing linear scaling.** Fig. 10 observes linear scaling between time vs. $m_i$ (top left), time vs. $n$ (bottom left), RAM vs. $m_i$ (top right) and RAM vs. $n$ (bottom right). Crucially, this helps ensure the practical applicability of our method in terms of implementation and execution.

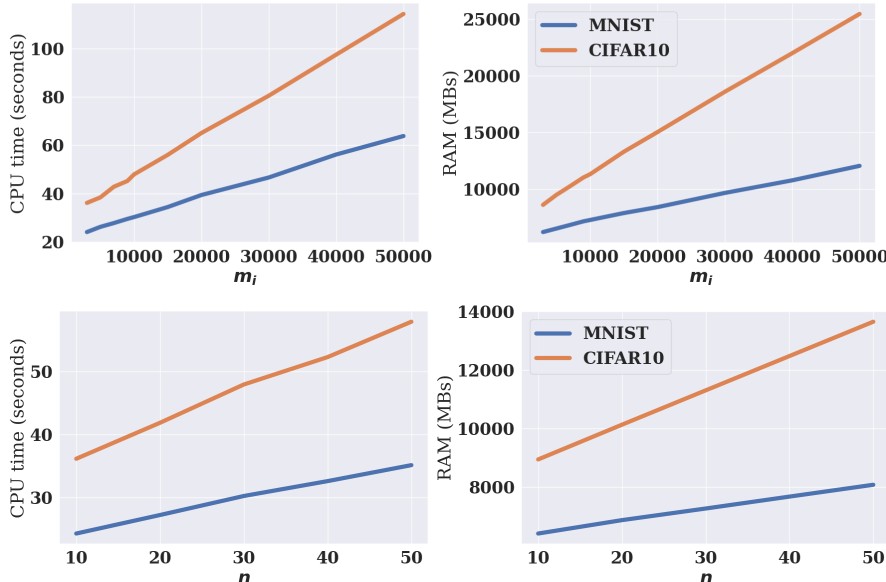

Figure 10: Top: time and peak memory vs. $m_i$ at $n = 30$; Bottom: time and peak memory vs. $n$ at $m = 10000$. MNIST or CIFAR10 denotes the dataset used.

**RAM, CUDA and CPU time results.** We include the detailed results for RAM, CUDA, and time on MNIST (Tables 19 to 21) and CIFAR10 (Tables 16 to 18), respectively.

Table 16: CUDA Memory in MBs for CIFAR10.

| $m_i \backslash n$ | 10 | 20 | 30 | 40 | 50 |
|---|---|---|---|---|---|
| 3000 | 140.598272 | 182.904320 | 222.904320 | 264.693248 | 304.847360 |
| 5000 | 141.556736 | 182.904320 | 222.904320 | 264.693248 | 304.847360 |
| 7000 | 143.563776 | 182.904320 | 222.904320 | 264.693248 | 304.847360 |
| 9000 | 144.983040 | 182.904320 | 222.904320 | 264.693248 | 304.847360 |
| 10000 | 145.862144 | 182.904320 | 222.904320 | 264.693248 | 304.847360 |
| 15000 | 150.258176 | 182.904320 | 222.904320 | 264.693248 | 304.847360 |
| 20000 | 155.048960 | 182.904320 | 222.904320 | 264.693248 | 304.847360 |
| 30000 | 164.999168 | 189.000192 | 222.904320 | 264.693248 | 304.847360 |
| 40000 | 173.798400 | 205.969920 | 238.375936 | 271.930368 | 304.847360 |
| 50000 | 182.904320 | 222.904320 | 264.693248 | 304.847360 | 346.482176 |

**Scalability comparison against DAVINZ.** The implementation of DAVINZ also includes an MMD computation (similar to our proposed method), but additionally linearly combined with a the neural tangent kernel (NTK)-based score. While in some cases (e.g., Tables 3 and 4), DAVINZ and our proposed method perform comparably, we highlight that our method is more scalable. The main reason is that the gradient computation from NTK in DAVINZ requires additional memory, specifically CUDA memory due to leveraging GPU for gradient computation. See Table 22 for a scalability experiment for DAVINZ with $n = 10$ on MNIST with a standard convolutional neural network used for all MNIST-related experiments in this work.

In contrast, under the same setting of $n = 10$ data vendors each with $m_i = 10000$ samples, our method requires less than 0.1 GBs of CUDA memory (see the fifth row, first column of Table 19. Note that we were not able to collect results for DAVINZ on more extensive settings due to hardware

Table 17: RAM in MBs for CIFAR10.

| $m_i \backslash N$ | 10 | 20 | 30 | 40 | 50 |
|---|---|---|---|---|---|
| 3000 | 7726.480469 | 8022.242188 | 8602.214844 | 8813.792969 | 9256.371094 |
| 5000 | 8036.699219 | 8683.101562 | 9489.097656 | 10134.582031 | 10780.066406 |
| 7000 | 8318.515625 | 9469.835938 | 10239.902344 | 11060.789062 | 11963.085938 |
| 9000 | 8628.960938 | 9815.738281 | 11016.425781 | 12092.296875 | 13129.046875 |
| 10000 | 8945.777344 | 10132.257812 | 11306.156250 | 12478.398438 | 13650.953125 |
| 15000 | 9444.472656 | 11187.527344 | 13286.566406 | 14993.132812 | 16758.398438 |
| 20000 | 10058.050781 | 12478.628906 | 15021.453125 | 17166.140625 | 19696.039062 |
| 30000 | 11374.105469 | 15036.773438 | 18574.199219 | 22034.144531 | 25492.121094 |
| 40000 | 12545.460938 | 17267.832031 | 21975.371094 | 26613.242188 | 31401.500000 |
| 50000 | 13877.929688 | 19598.089844 | 25435.593750 | 31254.929688 | 37299.406250 |

Table 18: CPU time in seconds for CIFAR10.

| $m_i \backslash N$ | 10 | 20 | 30 | 40 | 50 |
|---|---|---|---|---|---|
| 3000 | 40.075573 | 33.853308 | 36.099317 | 36.748012 | 38.443388 |
| 5000 | 32.896979 | 35.561858 | 38.386079 | 41.070452 | 44.552341 |
| 7000 | 34.791889 | 38.024787 | 42.796535 | 45.675799 | 50.420825 |
| 9000 | 37.090735 | 40.622674 | 45.164939 | 50.043854 | 55.225169 |
| 10000 | 36.123430 | 41.834278 | 47.907592 | 52.253957 | 57.867871 |
| 15000 | 38.382908 | 47.813846 | 56.015236 | 63.631425 | 72.990464 |
| 20000 | 42.548813 | 54.514572 | 65.067422 | 76.599092 | 89.003306 |
| 30000 | 46.957031 | 65.156033 | 80.534339 | 97.402354 | 116.374963 |
| 40000 | 52.663162 | 76.029116 | 97.542482 | 120.323142 | 144.956402 |
| 50000 | 59.434006 | 89.602017 | 114.438738 | 146.690117 | 173.507339 |

Table 19: CUDA memory for MNIST.

| $m_i \backslash N$ | 10 | 20 | 30 | 40 | 50 |
|---|---|---|---|---|---|
| 3000 | 95.339008 | 138.832384 | 178.832384 | 220.621312 | 260.775424 |
| 5000 | 97.097728 | 138.832384 | 178.832384 | 220.621312 | 260.775424 |
| 7000 | 98.855936 | 138.832384 | 178.832384 | 220.621312 | 260.775424 |
| 9000 | 101.214720 | 138.832384 | 178.832384 | 220.621312 | 260.775424 |
| 10000 | 101.693952 | 138.832384 | 178.832384 | 220.621312 | 260.775424 |
| 15000 | 105.895936 | 138.832384 | 178.832384 | 220.621312 | 260.775424 |
| 20000 | 110.686720 | 138.832384 | 178.832384 | 220.621312 | 260.775424 |
| 30000 | 119.679488 | 143.097856 | 178.832384 | 220.621312 | 260.775424 |
| 40000 | 130.443776 | 162.443776 | 195.455488 | 227.238400 | 260.775424 |
| 50000 | 138.832384 | 178.832384 | 220.621312 | 260.775424 | 302.410240 |

limitations (i.e., larger values for $n$ and $m_i$ leads to out-of-memory errors on our standard GPUs with 10 GBs of memory and the official implementation (Wu et al., 2022, Github repo) does not implement a way to take advantage of multiple GPUs (if available) to distribute the CUDA memory load.

Table 20: RAM in MBs for MNIST.

| $m_i \backslash N$ | 10 | 20 | 30 | 40 | 50 |
|---|---|---|---|---|---|
| 3000 | 6077.753906 | 6148.613281 | 6204.640625 | 6332.785156 | 6478.375000 |
| 5000 | 6093.554688 | 6284.605469 | 6517.027344 | 6718.703125 | 6934.570312 |
| 7000 | 6122.351562 | 6535.925781 | 6828.468750 | 7151.273438 | 7438.753906 |
| 9000 | 6206.800781 | 6694.765625 | 7148.257812 | 7543.253906 | 7876.445312 |
| 10000 | 6414.144531 | 6872.023438 | 7269.164062 | 7676.304688 | 8080.292969 |
| 15000 | 6665.367188 | 7210.253906 | 7880.492188 | 8484.843750 | 9024.613281 |
| 20000 | 6867.117188 | 7651.234375 | 8402.281250 | 9231.320312 | 9992.460938 |
| 30000 | 7205.328125 | 8409.937500 | 9653.914062 | 10725.601562 | 11970.367188 |
| 40000 | 7559.824219 | 9132.652344 | 10764.582031 | 12254.757812 | 13927.617188 |
| 50000 | 8096.402344 | 9891.027344 | 12040.925781 | 13953.308594 | 15842.777344 |

Table 21: CPU time in seconds for MNIST.

| $m_i \backslash N$ | 10 | 20 | 30 | 40 | 50 |
|---|---|---|---|---|---|
| 3000 | 45.675888 | 23.173012 | 24.010044 | 24.685510 | 25.532540 |
| 5000 | 22.843993 | 24.338142 | 26.162306 | 27.275437 | 28.386587 |
| 7000 | 23.884181 | 25.619524 | 27.735549 | 29.385311 | 31.042997 |
| 9000 | 23.949323 | 26.902086 | 29.450777 | 31.600939 | 33.862759 |
| 10000 | 24.259586 | 27.199071 | 30.215128 | 32.577134 | 35.111519 |
| 15000 | 28.153503 | 30.075059 | 34.379974 | 38.391948 | 42.446352 |
| 20000 | 27.595278 | 34.082700 | 39.362670 | 46.231271 | 49.121318 |
| 30000 | 30.255474 | 38.232212 | 46.626252 | 54.726982 | 63.040075 |
| 40000 | 33.875198 | 44.688145 | 56.146309 | 66.236627 | 77.427208 |
| 50000 | 35.978749 | 50.792472 | 63.822176 | 77.274825 | 92.584505 |

Table 22: Maximum CUDA, RAM and time for DAVINZ with $n = 10$ data vendors on MNIST with a standard convolutional neural network.

| $m_i$ | maximum CUDA (in MBs) | RAM (in MBs) | CPU time (in seconds) |
|---|---|---|---|
| 3000 | 4885.189 | 5201.132 | 50.419 |
| 5000 | 4885.189 | 6840.75 | 44.173 |
| 7000 | 4885.189 | 6843.371 | 46.229 |
| 9000 | 4885.189 | 6836.921 | 51.436 |
| 10000 | 5523.673 | 6843.773 | 50.491 |
| 15000 | 5523.673 | 6926.535 | 56.739 |
| 20000 | 7118.552 | 7358.734 | 68.144 |
| 30000 | 7118.552 | 8331.660 | 86.353 |
| 40000 | 7323.634 | 9154.773 | 107.132 |
| 50000 | 7323.634 | 10053.316 | 132.206 |

