# OpenReview forum: "Data Distribution Valuation with Incentive Compatibility"
_ICLR.cc/2024/Conference — Submitted to ICLR 2024_

### Official Review · Reviewer_uc12 · 2023-10-25

**Soundness:** 2 fair
**Presentation:** 3 good
**Contribution:** 2 fair
**Rating:** 3
**Confidence:** 3

**Summary:**

This paper proposes a method to assess the value of a data distribution (instead of a sampled dataset), and claims that this method satisfies incentive-compatibility (IC) in the sense that data providers have no incentives to mis-report their data.

The main idea of the method is the following: First, assume that each provider's distribution $P_i$ is a mixture of a reference (ground truth) distribution $P^*$ and a noise distribuiton $Q_i$. Then, the maximum mean discrepancy (MMD) $d(P_i, P^*$) between $P_i$ and the reference distribution $P^*$ is a value measure for $P_i$. Since $d(P_i, P^*)$ is unknown, the authors approximate it by $\hat d(D_i, D^*)$ using sampled datasets $D_i$, $D^*$. In addition, since the reference distribution $P^*$ is also unknown, the authors propose to use the average of all data providers' distributions as a proxy, $P_N$. Hence, the final metric is $\hat \nu(D_i) = \hat d(D_i, D_N)$, where $D_N$ is the aggregate of all providers' datasets.

The authors then prove that this metric $\hat \nu$ satisfies IC if: Roughly speaking, (1) $d(P_{-i}, P^*)$ is small (_the average distribution of providers other than $i$ is closed to the reference $P^*$_) or $d(\tilde P_i, P^*)$ is large (_the report of $i$ is far from $P^*$_) (from Corollary 1); (2) the size $m_i$ of each sampled dataset $D_i$ is large enough (from Theorem 1).

The authors then present some experimental results to support their theoretical claims.

**Strengths:**

1. The use of Huber model (each provider's distribution $P_i$ is a mixture of a reference distribution $P^*$ and a noise distribuiton $Q_i$) is natural and interesting.

2. The main problem -- measuring the value of a data distribution, not just a sampled dataset of limited size -- is well motivated as in the introduction.

**Weaknesses:**

Although the problem of data distribution valuation is well motivated, I don't think the authors' solution to this problem is satisfactory:

1. A main idea in the authors' solution is to use the MMD $\hat d(D_i, D_N)$ estimated from samples to approximate the true MMD $d(P_i, P^*)$. This is basically a law of large number, where good estimation can be obtained if the number of samples is large enough. However, the motivation of data distribution valuation is exactly the lack of samples. As written in the Introduction, "data vendors offer a preview in the form of a sample dataset to prospective buyers. Buyers use these sample datasets to decide whether they wish to pay for a full dataset or data stream." The key tension is how the buyers can evaluate the value of a full dataset given access to only a limited-size preview dataset. **This is at odd with the authors' approach of using many samples to estimate the value of the distribuiton.**

2. Another idea in the authors' solution is to use the average distribution $P_N$ of every one's distribution as the reference distribution $P^*$. This requires $P_N$ to be closed to $P^*$; in other words, the noisy parts $Q_i$ in everyone's distribution cancel out when averaged (Proposition 2). **This is a strong limitation.** As the authors point out (in page 17), it is possible that a majority of data providers are off from $P^*$ while only a few are close to $P^*$. In this case, the authors' solution does not work.

3. **My biggest concern is the definition of Incentive-Compatibility (Definition 1)**, which says that "a vendor with actual data distribution $P$ will not choose to report data from distribution $\tilde P$ where **w.l.o.g.** $d(P, P^*) < d(\tilde P, P^*)$". I don't understand why this is "without loss of generality". A vendor can definitely report data from a distribution $\tilde P$ that is closer to $P^*$ than the actual distribution $P$ is: $d(\tilde P, P^*) < d(P, P^*)$. In the standard definition of IC (like in [Chen et al 2020]), a player should be disincentivized to mis-report anything.

**Questions:**

**Question:**

In Section 6.2 (experiment about incentive compatibility), the authors let the vendors misreport $\tilde D_i$ by adding zero-mean Gaussian noise to the original data $D_i$:

(1) Why cannot the vendors use other form of misreporting?  Did you try other form of misreporting?

(2) And why does adding Gaussian noise ensures $d(\tilde P_{i'}, P^*) > d(P_{i'}, P^*)$ ?

**Suggestion:**

Maybe a better motivation/theme of this work should be "how to value heterogeneous datasets", instead of "how to value data distribution instead of dataset". As I argued in Weakness 1, the authors' approach is not about valuing data distributions using limited-size preview datasets; it is more about how to value heterogeneous datasets.

---

> ### Author Response · Authors · 2023-11-18
> **Author response (1/2)**
>
> We thank Reviewer uc12 for taking the time to review our paper and providing the detailed feedback, and for recognizing that our modeling approach is **natural** and **interesting** and our main studied problem is **well motivated**.
>
> ---
>
> We wish to provide the following clarifications.
>
> > However, the motivation of data distribution valuation is exactly the lack of samples. ... This is at odd with the authors' approach of using many samples to estimate the value of the distribuiton.
>
> We hope to highlight that while the constraint (i.e., lack of samples) may appear to be at odds with the proposed approach (i.e., using samples to estimate the value), this is an important motivation for the specific design choice namely MMD due to its efficient sample complexity.
>
> - [Better sample complexity.] The sample complexity of MMD (i.e., using sample MMD to estimate true MMD) is **efficient**, precisely $\mathcal{O}(1/ {\sqrt{m}})$ (where $m$ the size of the sample) and importantly, **independent of the dimension**. In contrast, the optimal transport (as discussed in Appendix B.1) has a sample complexity of $\mathcal{O} 1(1/n^{1/d})$ which suffers the *curse of dimensionality* (Genevay et al., 2019).
>
> - [Faster empirical convergence.] The benefit of this efficient sample complexity is demonstrated in an empirical comparison in Appendix C.3.1 where the sample MMD converges more quickly (to the true MMD), making MMD an more practically appealing choice (than other divergence) under the constraint that only a limited-size sample is available.
>
> ---
>
> > This requires $P_N$ to be close to $P^*$; in other words, the noisy parts $Q_i$ in everyone's distribution cancel out when averaged (Proposition 2).
>
> We wish to clarify that Proposition 2 does _not_ require the assumption that $P_N$ is close to $P^*$. Instead, Proposition 2 shows the effect of the distance of $P_N$ from $P^*$. Providing a precise characterization of how the heterogeneity in the vendors (i.e., the $Q_i$'s and the $\epsilon_i$'s) affects the valuation function, is a contribution, as it has not been previously obtained: The same design choice (of using $P_N$ as a majority-based reference instead of $P^*$) has been previously adopted by Chen et al. (2020); Tay et al. (2022); Wei et al. (2021), as elaborated in Appendix B. The key distinction is that they do *not* provide the error guarantee (i.e., Proposition 2).
>
> Indeed, because of the possibility that $P_N$ may be far from $P^*$, we have obtained results (in Appendix A.2.4) that are general and can be applicable to other alternative reference $P_{\text{ref}}$ (to $P_N$), should it become available by design or assumption. In this regard, these considerations do *not* have the limitation that $P_N$ must be close to $P^*$.
>
> ---
>
> > In the standard definition of IC (like in [Chen et al 2020]), a player should be disincentivized to mis-report anything.
>
> Please refer to the general comment "Updated definition for incentive compatibility and theoretical results (in Appendix A.3)".

---

> > ### Author Response · Authors · 2023-11-18
> > **Author response (2/2)**
> >
> > ---
> >
> > Q1
> > > Why cannot the vendors use other form of misreporting? Did you try other form of misreporting?
> >
> > Yes, the vendors can consider other form of mis-reporting and the corresponding empirical results are in Appendix C.3.3. In particular, these results are obtained w.r.t. a non-Huber setting, which is to specifically examine how our method performs outside of our assumed setting (i.e., Huber) and our method remains effective (albeit not always the best).
> >
> > ---
> >
> > Q2
> > > And why does adding Gaussian noise ensures $d(\tilde{P}_{i'}, P^*) > d(P_{i'}, P^*)$?
> >
> > Intuitively, it is because the data (e.g., MNIST images as ground truth $P^*$) are *not* (isotropic) Gaussian noise, so adding (isotropic) Gaussian noise makes the data "worse" in terms of a higher MMD to the ground truth.
> >
> > ---
> >
> > For the suggestion:
> >
> > We hope to clarify that our focus is to *characterize and precisely define a value for a data distribution*, motivated by the use-cases mentioned in Section 1. Our theoretical framework (built on the Huber model and the MMD) is proposed to enable theoretical analysis w.r.t. the definitions of the data distributions (e.g., Observation 1 and Eq. (2)). In other words, results such as Observation 1 and Eq. (2) are specifically targeting the valuation of data distribution.
> >
> > Then, our proposed sample-based approach (along with the results including Proposition 1, Theorem 1 and Proposition 2) is to observe the practical constraint that the analytic expression of the sampling distribution (or data distribution) is __not attainable__ and to ensure our theoretical framework can still be implemented in practice, _without_ the analytic expressions of $P_i$'s. In particular, Proposition 1 and Theorem 1 are specific designs for implementations to obtain actionable policies to compare the values of the data distributions.
> >
> > ---
> >
> > We hope our response has addressed your comments and helped raise your opinion of our work.
> >
> > **References**
> >
> > Aude Genevay, Lénaic Chizat, Francis Bach, Marco Cuturi and Gabriel Peyré. Sample Complexity of Sinkhorn Divergences. AISTATS, 2019.

---

> > ### Comment · Reviewer_uc12 · 2023-11-22
> > **IC definition is not good, even after the update**
> >
> > I read the updated definition of IC in Appendix A.3.  **It requires that the mis-reported distribution $\tilde P_i$ is close to the true distribution $P_i$ by at most a distance of $\kappa_i$**.  Under this requirement, the authors prove that the mechanism is $\kappa$-IC in Proposition 4.  In my opinion this is a "circular argument" that does not make much sense.  By directly requiring the agents to report something close to their true distribution by $\kappa$, then of course the mis-reporting score $\Upsilon(\tilde P_i)$ is at most $\kappa$-worse than the truthful reporting score $\Upsilon(P_i)$ by a triangle inequality.
> >
> > In contrast, the standard definition of IC in game theory never puts such requirement of the misreports.  An IC mechanism in the standard definition should guarantee that the agent gets a lower score than truthful reporting no matter what the agent reports (e.g., they can report some $\tilde P_i$ that is far away from $P_i$ but close to $P^*$, or any other distribution, and still gets a score less than reporting $P_i$).
> >
> > Obtaining incentive-compatibility in data valuation and elicitation is known to be a challenging problem in the literature.  If this paper's way to define and obtain IC is acceptable, then all previous works on this topic become worthless.  So I continue to recommend rejection.
> >
> > If I were the authors, I would remove the IC results and keep all the other results, and then resubmit to another conference.

---

> > > ### Author Response · Authors · 2023-11-23
> > > **Thank you for engaging with us in discussion!**
> > >
> > > We hope to clarify with the reviewer that the reviewer means (the assumption of) the mechanism is not standard, instead of the updated definition of IC itself is not standard, since the form of the updated definition follows from existing works (Balseiro et al., 2022; Balcan et al., 2019).
> > >
> > > Nevertheless, we do see that the notion of IC in this particular setting (i.e., where the truth $P_i$ is not necessarily the most valuable since it is not ground truth) may not be the most suitable, and will incorporate the reviewer's suggestion in our revision.
> > >
> > > We hope to clarify with the reviewer whether the other questions (other than IC) have been addressed, and thank the reviewer again for engaging with us!

---

> ### Author Response · Authors · 2023-11-22
> **Let us know if you have further questions**
>
> We wish to thank Reviewer uc12 for the taking the time to review our paper and providing the feedback, and hope that our response has clarified your raised questions and comments. Since the discussion period is drawing to the end, let us know if you have further questions and we are happy to clarify them before the discussion period ends.

---

### Official Review · Reviewer_s7hX · 2023-10-31

**Soundness:** 1 poor
**Presentation:** 1 poor
**Contribution:** 1 poor
**Rating:** 1
**Confidence:** 4

**Summary:**

The paper studies data valuation, focusing on proposing valuation metric that encourages truthful reporting of data vendors (incentive compatibility). The model assumes Huber classes of distribution, where each vendor $i$ holds a mixture $P_i$ of some arbitrary distribution (stochastic or adversarial) and some ground-truth distribution $P^*$, and presents a sample from it to the user. The user hopes to evaluate the quality of the distribution (presumably in terms of proximity to $P^*$) through the samples, when both $P_i$ and $P^*$ may be unknown. Further, the designed metric should encourage vendors to report truly i.i.d. samples from their distribution, instead of making up artificial samples. The proposed solution uses the idea of maximum mean discrepancy (MMD), applying an estimator of MMD on the samples. Theoretical analysis and empirical evaluation are both offered.

**Strengths:**

The problem studied in this paper is meaningful and interesting.

**Weaknesses:**

- [Quality] I find the setting studied in this paper slightly confusing. In particular, I am not sure if the author(s)' definition of IC is in line with my usual understanding: it looks like the definition of $\gamma$ -IC (Def. 1 and Cor. 1) depends on the choice of $\tilde{P}$ (instead of "for all $\tilde{P}$ in certain class"), which seems unusual. The approach in this paper is to propose a natural candidate for the metric, and then studies its IC property (as opposed to characterizing the class of IC designs) - there is nothing wrong with it, but I find it slightly misleading to refer to Cor. 1 as $\gamma$ -IC. I am quite certain that manipulating the dataset would quite often lead to better metrics. Moreover, the empirical results also seem insufficient in justifying the true IC (in the way I would define it) - the alternative distribution considered is overly specific and somewhat naive, and clearly cannot represent the class of all possible manipulations. Overall, I am not convinced of the correctness (or significance) of the main claim.
- [Clarity] Certain parts in the paper are confusing to read with slightly informal tone, primarily due to the lack of formal and rigorous mathematical statements. For instance, the wording of Def. 1 (and also the Problem Statement) is imprecise (and, to my previous point, likely nonstandard).

**Questions:**

- First and foremost, please verify if Def. 1 is consistent with the standard notion of IC.
- I am curious why the problem is phrased "data valuation"? I see nothing about pricing in this work, and to me "evaluation" seems like a better fit?

---

> ### Author Response · Authors · 2023-11-18
> **Author response**
>
> We thank Reviewer s7hX for taking the time to reviewer our paper and for finding that our studied problem is **meaningful and interesting**.
>
> ---
> We hope to address the comments as follows.
>
> [On quality/setting.]
>
>
> >  I am quite certain that manipulating the dataset would quite often lead to better metrics.
>
>
> Could the reviewer please confirm that by "manipulating the dataset leading to better metrics", it means the vendor performs some careful statistical modification of the dataset to improve its value?
>
>
> > the alternative distribution considered is overly specific and somewhat naive, and clearly cannot represent the class of all possible manipulations
>
> The "alternative distribution" is specified to be interpretable and intuitive, so that the empirical results can be understood and used to verify the theoretical results. A very complex "alternative distribution" can make the results and analysis difficult and thus not very useful in deriving insights.
>
> We note that we do _not claim to represent the class of all possible manipulations_, in particular our empirical settings.
>
> ---
>
> [On clarity.]
>
> Please find an updated formalization of the problem statement as follows:
>
> Denote the ground truth (i.e., optimal) distribution as $P^*$. For two data distributions $P, P'$, and the respective samples $D\sim P, D\sim P'$, design a function $\Upsilon(P) \mapsto \mathbb{R}$, and a function $\nu(D) \mapsto \mathbb{R}$, so that for a specified (given) $\epsilon_P > 0, \delta \in [0,1]$, we can derive the $\epsilon_D \geq 0$ for which the following holds with probability at least $1-\delta$:
> $ \nu(D) \geq \nu(D') + \epsilon_D \implies \Upsilon(P) \geq \Upsilon(P') + \epsilon_P \ . $
>
> Then, our theoretical results Proposition 1 and Theorem 1 are the solutions to this problem, with and without the assumption of $P^*$ being available, respectively.
>
> ---
> > First and foremost, please verify if Def. 1 is consistent with the standard notion of IC.
>
> Please refer to the general comment "Updated definition for incentive compatibility and theoretical results (in Appendix A.3)".
>
>
>
> > I am curious why the problem is phrased "data valuation"? I see nothing about pricing in this work, and to me "evaluation" seems like a better fit?
>
> We note that the term "data valuation" is a relatively common phrase in similar existing works (Ghorbani & Zou, 2019; Kwon & Zou, 2022, Amiri et al., 2022, Wang & Jia, 2023, Just et al., 2023, Wu et al., 2022, Bian et al. 2021, Jia et al. 2018; 2019, Yoon et al. 2020).
>
> As described in Section 1, one primary motivating application of this work (and these above existing works) is to use the value of data for the pricing of data in data marketplaces such as Datarade, Snowflake (Chen et al., 2022) (reference in main paper) and (Agarwal et al., 2019, Sim et al., 2022) (references below).
>
> ---
>
> We hope our response (and our updated definition of IC) have addressed your comments and helped improve your opinion of our work.
>
> **References**
>
> Agarwal, A., Dahleh, M., & Sarkar, T. (2019). A marketplace for data: An algorithmic solution. Proc. EC, 701–726.
>
> Rachael Hwee Ling Sim, Xinyi Xu and Bryan Kian Hsiang Low. (2022) Data Valuation in Machine Learning: “Ingredients”, Strategies, and Open Challenges. IJCAI, survey track.

---

> ### Author Response · Authors · 2023-11-22
> **Let us know if we our response has clarified your comments**
>
> We wish to thank Reviewer s7hX for the taking the time to review our paper and providing the feedback, and hope that our response has clarified your questions, in particular regarding the definition of IC. Since the discussion period is drawing to the end, let us know if you have further questions and we are happy to clarify them before the discussion period ends.

---

### Official Review · Reviewer_c1ZW · 2023-11-12

**Soundness:** 2 fair
**Presentation:** 3 good
**Contribution:** 1 poor
**Rating:** 5
**Confidence:** 4

**Summary:**

This paper studies the problem of evaluating datasets from heterogeneous distributions. The paper adopts a Huber model on the dataset, assuming each dataset can be contaminated by samples from some noise distribution for a small factor. The principal aims to elicit truthful datasets from the data providers, and at the same time, the payment also distinguishes dataset qualities. The paper proposes to use the negative Maximum Mean Discrepancy (MMD) as the value of a distribution and to use an estimator of the MMD as the payment to incentivize truthful reports.

**Strengths:**

The paper studies an interesting problem and is well-motivated.

**Weaknesses:**

* MMD as a valuation doesn’t reflect data distribution’s performance on a real decision problem / proper loss function. In contrast divergence-based valuations correspond to the value of dataset on a proper loss.

* I suggest the author compare the results to previous results in information elicitation. Specifically, this following paper seems relevant. It studies elicitation with sample access to the distribution to be elicited. Suppose the principal has sample access to ground truth distribution, and sets it as a reference, it seems like the approach in this paper could be applied.

    Frongillo, Rafael, Dhamma Kimpara, and Bo Waggoner. "Proper losses for discrete generative models."

* Suppose all N-1 distributions have the same deterministic bias, while only one distribution is the ground truth distribution. It seems like it is impossible in this case to discover or incentivize the most valuable dataset.

Minor comments:
* typo: Page 7, Subsequently, each Pi follows a Huber: Pi =(1−εi)P∗ +Q , missing εi before Q.

**Questions:**

* Impossibility results: as my comment 3 in Weakness, I wonder if the authors can prove impossibility results on discovering the truth. This can help justify the current approach in the paper.

* Why is this bad? In the paper: “To elaborate, there is no need to explicitly learn the distributions P, P ′ to obtain their distribution divergence (e.g., in Kullback-Leibler divergence-based valuations”. Explicitly learning the distribution doesn’t sound like a drawback to me. The selection of MMD is not well-motivated. Are there other metrics that can achieve similar results?

* Following the previous point, if the authors believe learning the explicit distribution is not feasible, is it possible to construct estimators of KL divergence from samples and directly estimate KL divergence similarly as the approach in this paper?

* Definition of IC is unclear to me, especially the part with wlog.

---

> ### Author Response · Authors · 2023-11-18
> **Author response (1/3)**
>
> We thank Reviewer c1ZW for taking the time to review our work and providing the constructive feedback and for recognizing that our studied problem is **interesting** and **well-motivated**. We wish to address the feedback and comments as follows,
>
> ---
>
> > MMD as a valuation doesn't reflect data distribution's performance on a proper loss function.
>
> In (Si et al., 2023), the opposite is argued that the MMD can be used to provide a proper scoring rule (i.e., proper loss), because the **_continuous ranked probability score_ (CRPS) is a proper scoring rule, which can be shown to be a form of MMD**. Specifically, in Section 2 (Background) of (Si et al., 2023):
>
> > __A popular proper loss is the continuous ranked probability score (CRPS)__, defined for
> two cumulative distribution functions (CDFs) $F$ and $G$ as $\text{CRPS}(F, G) = \int (F(y) − G(y))^2 \text{d}y$ ... The above CRPS can also be rewritten as an expectation relative to the distribution $F$:
> \begin{equation}
>  \text{CRPS}(F, y') =  -\frac{1}{2} \mathbb{E}_F |Y - Y'| + \mathbb{E}_F |Y -y'|
> \end{equation}
> where $Y, Y'$ are independent copies of a random variable distributed according to $F$.
>
> And then in the last paragraph of the Section 2:
>
> > A special case of IPMs is maximum mean discrepancy (MMD) (Gretton et al., 2008), in which $\mathcal{T} ={T : \Vert T \Vert_{\mathcal{H} } ≤ 1}$ is the set of functions with a bounded norm in a reproducing kernel Hilbert space (RKHS) with norm $\Vert \cdot \Vert_{\mathcal{H}}$; the __CRPS objective can be shown to be a form of MMD__ (Gretton et al., 2008).
>
> In this regard, we believe that MMD is indeed connected to a proper loss function.
>
> As to the other part of this comment:
>
> > MMD as a valuation doesn't reflect data distribution's performance on a real decision problem.
>
> Could the reviewer elaborate on the definition of _a real decision problem_ and its potential benefits?
>
>
> >  In contrast divergence-based valuations correspond to the value of dataset on a proper loss.
>
> A divergence-based valuation does __not necessarily correspond__ to the value of dataset on a proper loss, because the result recalled by Kimpara et al., (2023), namely Theorem D.0.1 (in their Appendix D) is specific to the __Bregman divergence__. For data valuation, some divergences such as $f$-divergence and optimal transport have been considered (as compared and discussed in our Appendix B.1). To the best our knowledge, these divergences are not directly equivalent to the Bregman divergence, so the result in (Kimpara et al., 2023) is not immediately applicable.
>
> ---
>
> > I suggest the author compare the results to previous results in information elicitation. Specifically, this following paper seems relevant.
>
>
> We thank the reviewer for the reference and will include in our revision. We wish to highlight the following __connection and distinctions__:
>
> In (Kimpara et al., 2023), Definition 3.3 (_Proper divergence_) is a key component used, and indeed MMD is __a stritcly proper divergence__. This can be proven using Theorem 5 of (Gretton et al., 2012), which ensures $\text{MMD}(p,q) = 0 \iff p=q$, and the fact that $\text{MMD}(p,q) \geq 0$ because MMD is equivalently a norm in the Hilbert space.
>
> The distinctions are in the _goal and a technical setting_ where the goal of Kimpara et al., (2023) is to __evaluate generative models__ in the __discrete__ setting while our goal is to __characterize the sampling distributions__ for data in the __continuous__ setting.
>
> Note that we are citing the conference version of the paper, which has a different ordering of the authors from that reference (arXiv-version) raised by the reviewer.
>
> > Suppose the principal has sample access to ground truth distribution, and sets it as a reference, it seems like the approach in this paper could be applied.
>
> This corresponds to Proposition 1 in Section 4 where we make this assumption of having access to the ground truth distribution. We subsequently relax this assumption in Section 5.
>
> ---
> > Suppose all N-1 distributions have the same deterministic bias, while only one distribution is the ground truth distribution. It seems like it is impossible in this case to discover or incentivize the most valuable dataset.
>
> We wish to highlight that our theoretical result on incentivization aims to provide a precise characterization of how the incentive compatibility is dependent on the $n$ distributions ($1$ for each of $n$ vendors), instead of trying to achieving the exact incentive guarantee for all possible cases. In the case described by the reviewer, our analysis **continues to apply**, the effect is that the effectiveness of the incentivization for the vendor with the ground truth distribution would be lower, and will decrease as the deterministic bias increases.

---

> > ### Author Response · Authors · 2023-11-18
> > **Author response (2/3)**
> >
> > ---
> >
> > Q1 **[Impossibility results.]** While it may be appealing to obtain such an impossibility result on discovering the (ground) truth, we wish to highlight that our focus is _not_ to discover the ground truth, but rather to present an analysis, when the (ground) truth is unknown, to precisely characterize how the level of heterogeneity (as in Huber) or deterministic bias (as in the reviewer's previous comment) affects the degree of incentivization guarantee. As a contrast, an impossibility result categorically stating that under certain conditions, the exact incentivization is impossible, may not provide additional such insight.
> >
> > That being said, we think an impossibility result is possible and worth exploring as a future direction. We will incorporate this comment in our revision.
> >
> > ---
> >
> > Q2 **[On explicitly learning $P,P'$.]** Explicitly learning $P,P'$ can introduce __additional and unnecessary__ assumptions because the learning may require an additional assumption (using a parametric form of the distributions e.g., Gaussian to learn $P,P'$, which assumes that $P,P'$ can be approximated well with Gaussian). Such assumptions limit the applicability of the method, especially for high-dimensional and complex $P,P'$.
> >
> > > The selection of MMD is not well-motivated.
> >
> > We highlight that the choice of __MMD has several benefits__ (as mentioned in Section 4, below Eq. (1)) including _non-parametric and easy to estimate_; _applicable to high-dimensional_ data; can provide _a interpretable result_ when applied to the Huber model; and satisfying the _triangle inequality_, which is key in deriving the theoretical results (e.g., Proposition 1).
> >
> > > Are there other metrics that can achieve similar results?
> >
> > There are indeed other metrics ($f$-divergence and optimal transport) considered for the similar problem, but to our knowledge, they are __not__ amenable to the same theoretical results (i.e., Theorem 1, Proposition 2) that MMD can achieve.
> >
> > There are also other __disadvantages of these alternatives__:
> >
> > - A disadvantage of $f$-divergence $D_f(P\Vert P')$ is _an additional assumption that $P$ must be absolutely continuous w.r.t. $P'$_ because otherwise $D_f(P\Vert P')$ becomes undefined due to a division by zero.
> >
> > - A disadvantage of optimal transport is the _practical difficulty in implementing the $\inf$ operation_ over all couplings of distributions, which has lead to an exsting work (Just et al., 2023) to implementing a surrogate instead of obtaining the actual distance directly.
> >
> >     The elaboration on these disadvantages along with other details is provided in a comparative discussion in Appendix. B.1.
> >
> > Furthermore, the empirical results in Appendix C.3.1 show that MMD enjoys __a better sample complexity when applied in practice__.
> >
> > ---
> >
> > Q3 **[Estimating KL directly.]**
> > We wish to  highlight that our focus is *not* designing new estimators for an existing divergence (e.g., KL). Nevertheless, there are some techniques, such as the $k$-nearest neighbors ($k$NN) estimation for KL (Wang et al., 2008).
> >
> > There are some disadvantages for adopting the KL and this sample-based estimation:
> > - (i) [**Direction of KL**] Since KL is not symmetric, the selection of the direction of KL is an additional question to consider during application;
> > - (ii) [**No finite sample guarantee**] The $k$NN estimation is only shown to provide asymptotic guarantees, instead of the **finite sample guarantee** for MMD (Lemma 1 in Appendix A.1), which is used to provide the theoretical results (e.g., Theorem 1). Furthermore, "the rate of convergence of this $k$NN estimator (for KL) can be arbitrarily slow depending on the distributions." (Section 1-B, Sriperumbudur et al., 2009);
> > - (iii) [**Difficult for high-dimensional data**] "The NN method becomes unreliable in a high-dimension" (Section IV, Wang et al., 2008) and "for increasing dimensionality of the data (in $\mathbb{R}^d$), the method (NN estimator) is increasingly difficult to implement." (Section 1-B, Sriperumbudur et al., 2009), while MMD is designed to be suitable to high-dimensional data (Gretton et al. 2008).
> >
> > ---
> >
> > Q4 **[Definition of IC.]**
> > Please refer to the general comment "Updated definition for incentive compatibility and theoretical results (in Appendix A.3)".
> >
> > We hope our response has clarified your questions and helped improved your opinion of our work.

---

> > > ### Author Response · Authors · 2023-11-18
> > > **Author response (3/3)**
> > >
> > > **References**
> > >
> > > Phillip Si, Zeyi Chen, Subham Sekhar Sahoo, Yair Schiff and Volodymyr Kuleshov. Semi-Autoregressive Energy Flows: Exploring Likelihood-Free Training of Normalizing Flows, ICML 2023.
> > >
> > > Arthur Gretton, Karsten Borgwardt, Malte Rasch, Bernhard Schölkopf and Alexander Smola. A Kernel Method for the Two-Sample Problem, 2008.
> > >
> > > Dhamma Kimpara, Rafael Frongillo and Bo Waggoner. Proper Losses for Discrete Generative Models, ICML 2023.
> > >
> > > Qing Wang, Sanjeev R. Kulkarni and Sergio Verdú. "Divergence estimation for multidimensional densities via k-nearest-neighbor distances." IEEE Transactions on Information Theory (55)5, 2009: 2392-2405.
> > >
> > > Bharath K. Sriperumbudur, Kenji Fukumizu, Arthur Gretton, Bernhard Sch¨olkopf and Gert R. G. Lanckriet. On Integral Probability Metrics, φ-Divergences and Binary Classification, 2009.

---

> > ### Comment · Reviewer_c1ZW · 2023-11-22
> >
> > Thanks the authors for the clarification. I'd like to add that my comment on proper loss is more of a discussion, not a key weakness. The framework in this paper is two steps: 1) setting a reference point, either the ground truth distribution, or the average of all reported distributions; 2) estimating the divergence metric on the samples. The paper considers eps-IC, so the main difficulty is to select a divergence with good sample complexity, and there's no need for a proper loss.
> >
> > The reason I mentioned proper loss is, a proper loss captures the decision loss from inaccurate data distribution. Suppose the principal faces a decision problem with an uncertain payoff-relevant state which is drawn from the ground truth distribution. The principal uses the data distribution to predict the payoff-relevant state. My question is how does she bound the loss from using inaccurate data distribution. For any decision problem, there's a proper loss calculating this loss.
> >
> > I'm not sure if the reference you provided shows a proper loss can be constructed with MMD. It seems like it's saying CRPS is proper, and it's a special case of MMD.
> >
> > Talking about exact IC, the paper I mentioned seems to solve the problem, assuming reference as the ground truth distribution. I assume their approach generalize to eps-IC when using the average of reports as a reference.

---

> > > ### Author Response · Authors · 2023-11-22
> > > **Thank you for engaging with us and clarifying the point on proper loss is not a key weakness**
> > >
> > > Thank you for engaging with our discussion, in particular for clarifying that the point on proper loss is *not* meant as a key weakness. We hope to provide the following clarifications:
> > >
> > > > The paper considers eps-IC, so the main difficulty is to select a divergence with good sample complexity, and there's no need for a proper loss.
> > >
> > > Your understanding of our approach (i.e., first selecting a referece and then using a sample-based estimation) and the implications (i.e., no need to explicitly construct a proper loss) is correct.
> > >
> > >
> > > > My question is how does she bound the loss from using inaccurate data distribution.
> > >
> > > The "bound(ing) the loss" can be obtained from Proposition 2, which provides the error from using this "inaccurate data distribution" instead of the ground truth.
> > >
> > > > I assume their approach generalize to eps-IC when using the average of reports as a reference.
> > >
> > > Your idea (about the mentioned paper) is indeed similar to our proposed approach, which utilizes the Proposition 2 (which bounds the loss or error from using the inaccurate data distribution), in order to derive the eps-IC. This is precisely to account for the practical situation where the ground truth is unavailable (but the average of distributions is naturally available).
> > >
> > > We thank Reviewer c1ZW for engaging in further discussion with us. We hope that our response helped clarify that our proposed approach is effective in solving the problem, without having to explicitly relying on a proper loss, and that it helped improve your opinion of our work.

---

> ### Author Response · Authors · 2023-11-22
> **Let us know if you have further questions**
>
> We wish to thank Reviewer c1ZW for the taking the time to review our paper and providing the feedback, and hope that our response has clarified your comments and questions. Since the discussion period is drawing to the end, let us know if you have further questions and we will be happy to clarify them before the discussion period ends.

---

### Author Response · Authors · 2023-11-18
**Updated definition for incentive compatibility and theoretical results (in Appendix A.3)**

We wish to thank all the reviewers for taking the time to review our paper and providing useful feedback. Based on the reviews, we have updated our definition on incentive compatibility and also the corresponding theoretical results, already included in Appendix A.3 in the updated pdf. We include these in Appendix A.3 instead of directly modifying the main paper so that it is easier to refer to these results together.

The updated definition and results are recalled here for easy reference:

**Definition 3** [$\gamma$-incentive compatibility]:
The valuation function $\Upsilon$ is $\gamma$-incentive compatible if:
$ \Upsilon(P_i; \\{P_{i'};i'\in -i \\} , \cdot) \geq \Upsilon (\tilde{P}\_i ; \\{P_{i'};i'\in -i \\} , \cdot) - \gamma $
where $\tilde{P}_i$ denotes the misreported version of $P_i$ by vendor $i$.


Here the value for vendor $i$ is the utility for $i$, which is a more commonly adopted terminology in related literature.

We introduce an additional $\kappa_i \geq 0$ as the (maximum) degree of misreporting by vendor $i$: $\tilde{P}_i \in \\{Q; d(Q, P_i) \leq \kappa_i \\}$ and denote $\kappa := \max_i \kappa_i\ .$ The corresponding results are Propositions 4 & 5 (in Appendix A.3, and their proofs provided therein):

**Proposition 4**:
$\Upsilon(P_i):= -d(P_i, P^*) $ from Eq.(1) is $\kappa$-IC.

**Proposition 5**:
$\hat{\Upsilon}:=-d(P_i, P_N) $ from Eq.(3) is $\gamma_{\hat{\Upsilon}}$-IC where $ \gamma_{\hat{\Upsilon}} := \max_i \kappa_i \frac{m_{-i}}{m_N}\ $.

We hope that this updated definition is more aligned with the standard definition approximate IC in existing literature, and the corresponding results can make our contribution clearer.

---

### Meta-Review · Area_Chair_UMcC · 2023-12-12

**Metareview:**

All reviewers raised several major issues regarding clarity and correctness of the paper and the paper requires a large revision and a new round of reviews before acceptance.

**Justification For Why Not Higher Score:**

Every reviewer was uniformly negative

**Justification For Why Not Lower Score:**

N/A

---

### Decision · Program_Chairs · 2024-01-16

Reject